# Learning Distributions of Complex Fluid Simulations with Diffusion Graph Networks

**Mario Lino**[1]    **Tobias Pfaff**[2]    **Nils Thuerey**[1]
[1]Technical University of Munich    [2]Google DeepMind

## Abstract

Physical systems with complex unsteady dynamics, such as fluid flows, are often poorly represented by a single mean solution. For many practical applications, it is crucial to access the full distribution of possible states, from which relevant statistics (e.g., RMS and two-point correlations) can be derived. Here, we propose a graph-based latent diffusion (or alternatively, flow-matching) model that enables direct sampling of states from their equilibrium distribution, given a mesh discretization of the system and its physical parameters. This allows for the efficient computation of flow statistics without running long and expensive numerical simulations. The graph-based structure enables operations on unstructured meshes, which is critical for representing complex geometries with spatially localized high gradients, while latent-space diffusion modeling with a multi-scale GNN allows for efficient learning and inference of entire distributions of solutions. A key finding is that the proposed networks can accurately learn full distributions even when trained on incomplete data from relatively short simulations. We apply this method to a range of fluid dynamics tasks, such as predicting pressure distributions on 3D wing models in turbulent flow, demonstrating both accuracy and computational efficiency in challenging scenarios. The ability to directly sample accurate solutions, and capturing their diversity from short ground-truth simulations, is highly promising for complex scientific modeling tasks.

## 1 Introduction

Numerically solving partial differential equations is essential in many scientific and engineering fields (Karniadakis & Sherwin, 2013), with fluid dynamics being of particular interest across a wide range of disciplines (Verma et al., 2018; Kochkov et al., 2021). Nevertheless, the high computational cost often limits the practical application of these methods and the exploration of large parameter spaces. While advances in deep learning have produced promising surrogate models, these typically focus on predicting mean flows (Lino et al., 2023) or trajectories (Kim et al., 2019; Stachenfeld et al., 2021). For long trajectories, such methods often suffer from instability issues (Lippe et al., 2024). Besides, in many applications, the primary interest is not in individual trajectories or mean flows but in the statistical properties and probability distributions of unsteady flow fields in statistical equilibrium (Pope, 2000). For instance, RMS fluctuations must be accounted for in the design of airfoils or when deciding the relative placement of aerodynamic components (Caros et al., 2022; Jané-Ippel et al., 2023), and turbulence research inherently revolves around flow field statistics.

While specialized techniques exist for certain applications – such as unsteady Reynolds-averaged simulations, which approximate the mean flow (Alfonsi, 2009; Pope, 2004) – obtaining accurate distributions and statistics in the general case still requires extensive, time-consuming simulations. Often, long simulations are needed, and in turbulent flows, very small time-steps are necessary. Additionally, a significant portion of compute time is spent during the warm-up phase, before reaching statistical equilibrium (Moin & Kim, 1982), and in complex systems, such as weather simulations, reliable results require running ensembles of perturbed simulations (Maher et al., 2021).

To address this, we leverage a diffusion model that directly learns the *distribution* of equilibrium flow states, enabling efficient sampling from the flow state distribution and allowing for the inexpensive computation of any desired distributional metric. Unlike prevalent grid-based diffusion models (Song et al., 2021; Lienen et al., 2024), our approach employs a graph-based representation, operating directly on meshes. This allows for the accurate representation of complex geometries

and the adaptive allocation of resolution (Pfaff et al., 2021) – a crucial advantage for complex 3D scenarios, which would otherwise require computationally expensive operations on a dense 3D grid.

However, integrating graph neural networks (GNNs) with diffusion models for large systems presents challenges, particularly due to limitations in the efficiency of message passing for propagating features across the graph, which can hinder global denoising. To address this, we introduce an efficient *multi-scale GNN architecture* to model the denoising transitions and propose working on a *compressed latent mesh*. Operating in this latent space not only reduces inference time but also mitigates the introduction of undesired high-frequency noise in the solutions. Additionally, we find that the underlying distributions can be accurately learned from short simulations, even when these simulations lack sufficient diversity to fully represent the flow statistics. We demonstrate that our approach can accurately capture full distributions where other methods, such as Gaussian mixture models or variational autoencoders (VAEs), suffer from noise and mode collapse. In addition, it exhibits strong generalization and can be significantly more efficient than numerical simulations.

## 2 RELATED WORK

Numerous deep learning algorithms have been proposed to address scientific problems (Guest et al., 2018; Jumper et al., 2021; Pathak et al., 2022), with fluid flow problems receiving considerable attention within this field (Morton et al., 2018; Bar-Sinai et al., 2019; Lino et al., 2023). At the same time, probabilistic models, such as Gaussian mixture models and generative adversarial networks (GANs) (Goodfellow et al., 2014), have enabled the modeling of probability distributions over plausible states of physical systems based on collections of system snapshots (Maulik et al., 2020; Drygala et al., 2022; Kim & Lee, 2020). Recently, denoising diffusion probabilistic models (DDPMs) have emerged as a powerful class of probabilistic models, initially applied to image generation (Ho et al., 2020; Nichol & Dhariwal, 2021), where GANs have struggled to capture the full diversity of the data distribution (Dhariwal & Nichol, 2021). These models have found significant utility in physical modeling as well, with applications such as flow field super-resolution and reconstruction (Shu et al., 2023; Li et al., 2023), uncertainty estimation in under-resolved simulations (Liu & Thuerey, 2024), and enhancing the long-term stability of autoregressive rollouts (Lippe et al., 2024; Rühling Cachay et al., 2024; Kohl et al., 2024). More closely related to our work, Gao et al. (2024) and Lienen et al. (2024) leveraged DDPMs to model the probability distribution of fully-developed flows.

However, most of these studies relied on CNNs to parameterize the denoising process, which generally restricts system representation to Cartesian grids. GNNs, on the other hand, offer greater flexibility and allow for the discretization of systems using unstructured meshes (Pfaff et al., 2021; Lino et al., 2022). While GNNs have shown efficacy in modeling steady and unsteady simulations involving complex geometries and uneven spatial gradients (Belbute-Peres et al., 2020; Lam et al., 2023), their use in combination with DDPMs remains relatively unexplored.

Recent efforts have applied DDPMs to graph structures for tasks such as molecule synthesis (Xu et al., 2022; Hoogeboom et al., 2022; Trippe et al., 2023; Vignac et al., 2023), protein generation (Wu et al., 2024), and inverse protein folding (Yi et al., 2024). More aligned with our work, Wen et al. (2023) combined DDPMs and GNNs to forecast the probability distribution of future system states. Their approach was applied to traffic and air quality prediction, where the system's evolution was conditioned on past states and the underlying graph structure. However, their work applied data compression only along the temporal dimension, neglecting – along with previous work – spatial compression. As a result, these approaches were restricted to small graphs. In contrast, our work targets systems with thousands of nodes in both 2D and 3D. This required modeling the denoising process by multi-scale GNNs. These typically follow a U-Net like architecture (Gao & Ji, 2019), with specialized pooling and unpooling layers (Lino et al., 2022). Additionally, by shifting the generative problem to a compressed latent space, we let the model focus on more meaningful features, while any residual noise in the generated samples is reduced when transforming them back to physical space, unlike previous GNN-based DDPMs. We also address a promising, yet overlooked, topic: extrapolating the full distribution of system states from incomplete simulations.

## 3 LEARNING TO SAMPLE FROM DISTRIBUTIONS OF SIMULATIONS

We propose *Diffusion Graph Networks* (DGNs), a model that learns the probability distribution of dynamical states of physical systems, defined by their discretization mesh and their physical

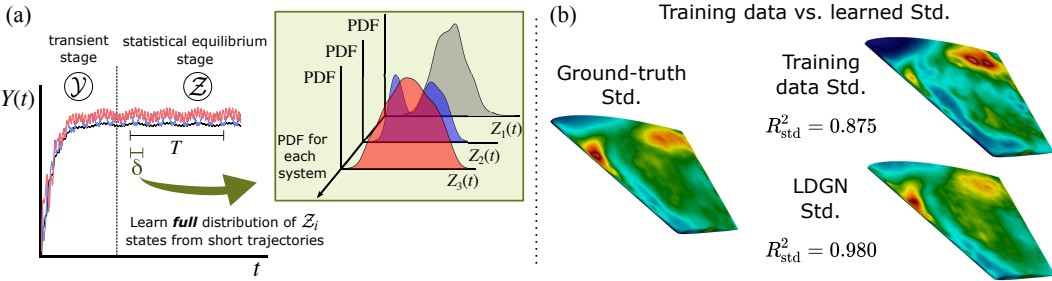

Figure 1: (a) We learn the probability distribution of the systems' converged states provided only a short trajectory of length $\delta << T$ per system. (b) A preview of our turbulent wing experiment. The distribution learned by our LDGN model accurately captures the variance of all states (bottom right), despite seeing only an incomplete distribution for each wing during training (top right).

parameters, by applying the DDPM framework to the mesh nodes. Additionally, we introduce a second model variant, the *Latent DGN* (LDGN), which operates in a pre-trained semantic latent space rather than directly in the physical space.[1]

We represent the system's geometry using a mesh with nodes $\mathcal{V}_M$ and edges $\mathcal{E}_M$, where each node $i$ is located at $\boldsymbol{x}_i$. The system's state at time $t$, $\boldsymbol{Y}(t)$, is defined by $F$ continuous fields sampled at the mesh nodes: $\boldsymbol{Y}(t) := \{\boldsymbol{y}_i(t) \in \mathbb{R}^F \mid i \in \mathcal{V}_M\}$, where $\boldsymbol{y}_i(t) \equiv \boldsymbol{y}(\boldsymbol{x}_i, t)$. Simulators evolve the system through an infinite sequence of states, $\mathcal{Y} = \{\boldsymbol{Y}(t_0), \boldsymbol{Y}(t_1), \dots, \boldsymbol{Y}(t_n), \dots\}$, starting from an initial state $\boldsymbol{Y}(t_0)$. We assume that after an initial transient phase, the system reaches a statistical equilibrium (see Figure 1a). In this stage, statistical measures of $\boldsymbol{Y}$, computed over sufficiently long time intervals, are time-invariant, even if the dynamics display oscillatory or chaotic behavior (Wilcox, 1998). The states in the equilibrium stage, $\mathcal{Z} \subset \mathcal{Y}$, depend only on the system's geometry and physical parameters, and not on its initial state.

In many engineering applications, such as aerodynamics and structural vibrations, the primary focus is not on each individual state along the trajectory, but rather on the statistics that characterize the system's dynamics (e.g., mean, RMS, two-point correlations). However, simulating a trajectory of converged states $\mathcal{Z}$ long enough to accurately capture these statistics can be very computationally expensive, especially for real-world problems involving 3D chaotic systems. To address this, we aim to directly sample converged states $\boldsymbol{Z}(t) \in \mathcal{Z}$ without simulating the initial transient phase. Subsequently, we can analyze the system's dynamics by drawing multiple samples.

Given a dataset of short trajectories from $N$ systems, $\mathfrak{Z} = \{\mathcal{Z}_1, \mathcal{Z}_2, ..., \mathcal{Z}_N\}$, our goal is to learn a probabilistic model of $\mathfrak{Z}$ that enables sampling of a converged state $\boldsymbol{Z}(t) \in \mathcal{Z}$, conditioned on the system's mesh, boundary conditions, and physical parameters. Crucially, this model must capture the underlying probability distributions even when trained on trajectories that are too short to fully characterize their individual statistics. Although this is an ill-posed problem, we hypothesize that, given sufficient training trajectories, it is possible to uncover their statistical correlations and shared patterns, enabling interpolation across the condition space.

## 3.1 DIFFUSION GRAPH NETWORKS

We use the DDPM framework (Ho et al., 2020; Nichol & Dhariwal, 2021) to generate states $\boldsymbol{Z}(t)$ by denoising a sample $\boldsymbol{Z}^R \in \mathbb{R}^{|\mathcal{V}_M| \times F}$ drawn from an isotropic Gaussian distribution. The system's conditional information is encoded in a directed graph $\mathcal{G} := (\mathcal{V}, \mathcal{E})$, where $\mathcal{V} \equiv \mathcal{V}_M$ and the mesh edges $\mathcal{E}_M$ are represented as bi-directional graph edges $\mathcal{E}$. Node attributes $\boldsymbol{V}_c = \{\boldsymbol{v}_i^c \mid i \in \mathcal{V}\}$ and edge attributes $\boldsymbol{E}_c = \{\boldsymbol{e}_{ij}^c \mid (i, j) \in \mathcal{E}\}$ encode the conditional features, including the relative positions between adjacent node, $\boldsymbol{x}_j - \boldsymbol{x}_i$. Domain-specific details on the node and edge encodings can be found in Appendix C and Table 4.

In the *diffusion* (or *forward*) process, node features from $\boldsymbol{Z}^1 \in \mathbb{R}^{|\mathcal{V}| \times F}$ to $\boldsymbol{Z}^R \in \mathbb{R}^{|\mathcal{V}| \times F}$ are generated by sequentially adding Gaussian noise: $q(\boldsymbol{Z}^r | \boldsymbol{Z}^{r-1}) = \mathcal{N}(\boldsymbol{Z}^r; \sqrt{1 - \beta_r} \boldsymbol{Z}^{r-1}, \beta_r \mathbf{I})$,

---

[1]Code is available at https://github.com/tum-pbs/dgn4cfd.

where $\beta_r \in (0, 1)$, and $Z^0 \equiv Z(t)$. Any $\boldsymbol{Z}^r$ can be sampled directly via:

$$\boldsymbol{Z}^r = \sqrt{\bar{\alpha}_r}\boldsymbol{Z}^0 + \sqrt{1 - \bar{\alpha}_r}\boldsymbol{\epsilon}, \tag{1}$$

with $\alpha_r := 1 - \beta_r$, $\bar{\alpha}_r := \prod_{s=1}^{r} \alpha_s$ and $\boldsymbol{\epsilon} \sim \mathcal{N}(\mathbf{0}, \mathbf{I})$ (Ho et al., 2020). The denoising process removes noise through learned Gaussian transitions: $p_\theta(\boldsymbol{Z}^{r-1}|\boldsymbol{Z}^r) = \mathcal{N}(\boldsymbol{Z}^{r-1}; \boldsymbol{\mu}_\theta^{r-1}, \boldsymbol{\Sigma}_\theta^{r-1})$, where the mean and variance are parameterized as (Nichol & Dhariwal, 2021):

$$\boldsymbol{\mu}_\theta^{r-1} = \frac{1}{\sqrt{\alpha_r}}\left(\boldsymbol{Z}^r - \frac{\beta_r}{\sqrt{1 - \bar{\alpha}_r}}\boldsymbol{\epsilon}_\theta^r\right), \qquad \boldsymbol{\Sigma}_\theta^{r-1} = \exp\left(\mathbf{v}_\theta^r \log\beta_r + (1 - \mathbf{v}_\theta^r)\log\tilde{\beta}_r\right), \tag{2}$$

with $\tilde{\beta}_r := (1 - \bar{\alpha}_{r-1})/(1 - \bar{\alpha}_r)\beta_r$. Here, $\boldsymbol{\epsilon}_\theta^r \in \mathbb{R}^{|\mathcal{V}| \times F}$ predicts the noise $\boldsymbol{\epsilon}$ in equation (1), and $\mathbf{v}_\theta^r \in \mathbb{R}^{|\mathcal{V}| \times F}$ interpolates between the two bounds of the process' entropy, $\beta_r$ and $\tilde{\beta}_r$.

DGNs predict $\boldsymbol{\epsilon}_\theta^{r-1}$ and $\mathbf{v}_\theta^{r-1}$ using a message-passing-based GNN (Battaglia et al., 2018). This takes $\boldsymbol{Z}^r$ as input, and it is conditioned on graph $\mathcal{G}$, its node and edge features, and the diffusion step $r$:

$$[\boldsymbol{\epsilon}_\theta^{r-1}, \mathbf{v}_\theta^{r-1}] \leftarrow \text{DGN}_\theta(\boldsymbol{Z}^r, \mathcal{G}, \boldsymbol{V}_c, \boldsymbol{E}_c, r). \tag{3}$$

We train the DGN using the loss function in equation (14). The full denoising process requires $R$ evaluations of the DGN to transition from $\boldsymbol{Z}^R$ to $\boldsymbol{Z}^0$, though more efficient sampling techniques exist (Nichol & Dhariwal, 2021; Song et al., 2021).

The DGN follows the widely used encoder-processor-decoder GNN architecture (Sanchez-Gonzalez et al., 2020; Pfaff et al., 2021). In addition to the node and edge encoders, our encoder includes a diffusion-step encoder, which generates a vector $\boldsymbol{r}_{\text{emb}} \in \mathbb{R}^{F_{\text{emb}}}$ that embeds the diffusion step $r$. The node encoder processes the conditional node features $\boldsymbol{v}_i^c$, alongside $\boldsymbol{r}_{\text{emb}}$. Specifically, the diffusion-step encoder and the node encoder operate as follows:

$$\boldsymbol{r}_{\text{emb}} \leftarrow \phi \circ \text{LINEAR} \circ \text{SINEMB}(r), \quad \boldsymbol{v}_i \leftarrow \text{LINEAR}\left([\phi \circ \text{LINEAR}(\boldsymbol{v}_i^c) \mid \boldsymbol{r}_{\text{emb}}]\right), \quad \forall i \in \mathcal{V}, \tag{4}$$

where $\phi$ denotes the activation function and SINEMB is the sinusoidal embedding function (Vaswani et al., 2017). The edge encoder applies a linear layer to the conditional edge features $\boldsymbol{e}_{ij}^c$. The encoded node and edge features are $\mathbb{R}^{F_h}$-dimensional vectors ($F_{\text{emb}} = 4 \times F_h$). We condition each message-passing layer on $r$ by projecting $\boldsymbol{r}_{\text{emb}}$ to an $F_h$-dimensional space and adding the result to the node features before each of these layers – i.e., $\boldsymbol{v}_i \leftarrow \boldsymbol{v}_i + \text{LINEAR}(\boldsymbol{r}_{\text{emb}})$. Details on message passing can be found in Appendix B.1.

Previous work on graph-based diffusion models has used sequential message passing to propagate node features across the graph (Hoogeboom et al., 2022; Wen et al., 2023). However, this approach fails for large-scale fluid flows, such as the ones studied in this paper, as denoising of global features becomes bottlenecked by the reach of message passing (Section 5). To address this, we adopt a multi-scale GNN for the processor, applying message passing on $\mathcal{G}$ and multiple coarsened versions of it in a U-Net fashion (Lino et al., 2022; Cao et al., 2023). This design leverages the U-Net's effectiveness in removing both high- and low-frequency noise (Si et al., 2023). To obtain each lower-resolution graph from its higher-resolution counterpart, we use Guillard's coarsening algorithm (Guillard, 1993), originally developed for fast mesh coarsening in CFD applications. As in the conventional U-Net, pooling and unpooling operations, now based on message passing, are used to transition between higher- and lower-resolution graphs. These operations are detailed in Appendix B.3.

## 3.2 LATENT DIFFUSION GRAPH NETWORKS

Inspired by latent diffusion models for CNNs (Rombach et al., 2022), we extend the DGN framework to operate in a lower-dimensional graph-based representation that is perceptually equivalent to $\mathfrak{Z}$. This space is defined as the latent space of a Variational Graph Auto-Encoder (VGAE) trained to reconstruct $\boldsymbol{Z}(t)$. We refer to a DGN trained on this latent space as a Latent DGN (LDGN).

In this configuration, the VGAE captures high-frequency information (e.g., spatial gradients and small vortices), while the LDGN focuses on modeling mid- to large-scale patterns (e.g., the wake and vortex street). By decoupling these two tasks, we aim to simplify the generative learning process, allowing the LDGN to concentrate on more meaningful latent representations that are less

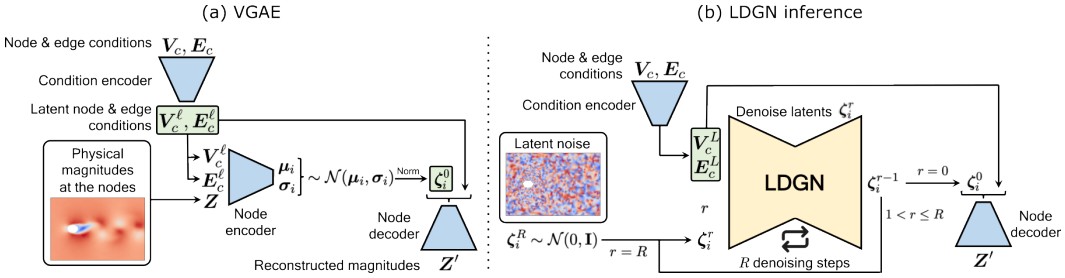

Figure 2: (a) Our VGAE consists of a condition encoder, a (node) encoder, and a (node) decoder. The multi-scale latent features from the condition encoder serve as conditioning inputs to both the encoder and the decoder. (b) During LDGN inference, Gaussian noise is sampled in the VGAE latent space and, after multiple denoising steps conditioned on the low-resolution outputs from the VGAE's condition encoder, transformed into the physical space by the VGAE's decoder.

sensitive to small-scale fluctuations. Additionally, during inference, the VGAE's decoder helps remove residual noise from the samples generated by the LDGN. This approach significantly reduces sampling costs since the LDGN operates on a smaller graph rather than directly on $\mathcal{G}$.

For our VGAE, we propose an encoder-decoder architecture with an additional condition encoder to handle conditioning inputs (Figure 2a). The condition encoder processes $\boldsymbol{V}_c$ and $\boldsymbol{E}_c$, encoding these into latent node features $\boldsymbol{V}_c^\ell$ and edge features $\boldsymbol{E}_c^\ell$ across $L$ graphs $\{\mathcal{G}^\ell := (\mathcal{V}^\ell, \mathcal{E}^\ell) \mid 1 \leq \ell \leq L\}$, where $\mathcal{G}^1 \equiv \mathcal{G}$ and the size of the graphs decreases progressively, i.e., $|\mathcal{V}^1| > |\mathcal{V}^2| > \cdots > |\mathcal{V}^L|$. This transformation begins by linearly projecting $\boldsymbol{V}_c$ and $\boldsymbol{E}_c$ to a $F_{\mathrm{ae}}$-dimensional space and applying two message-passing layers to yield $\boldsymbol{V}_c^1$ and $\boldsymbol{E}_c^1$. Then, $L-1$ encoding blocks are applied sequentially:

$$\left[\boldsymbol{V}_c^{\ell+1}, \boldsymbol{E}_c^{\ell+1}\right] \leftarrow \text{MP} \circ \text{MP} \circ \text{GRAPHPOOL}\left(\boldsymbol{V}_c^\ell, \boldsymbol{E}_c^\ell\right), \quad \text{for } l = 1, 2, \ldots, L-1, \tag{5}$$

where MP denotes a message-passing layer and GRAPHPOOL denotes a graph-pooling layer (see the diagram on Figure 9a).

The encoder produces two $F_L$-dimensional vectors for each node $i \in \mathcal{V}^L$, the mean $\boldsymbol{\mu}_i$ and standard deviation $\boldsymbol{\sigma}_i$ that parametrize a Gaussian distribution over the latent space. It takes as input a state $\boldsymbol{Z}(t)$, which is linearly projected to a $F_{\mathrm{ae}}$-dimensional vector space and then passed through $L-1$ sequential down-sampling blocks (message passing + graph pooling), each conditioned on the outputs of the condition encoder (Figure 9b):

$$\boldsymbol{V} \leftarrow \text{GRAPHPOOL} \circ \text{MP} \circ \text{MP}\left(\boldsymbol{V} + \text{LINEAR}\left(\boldsymbol{V}_c^\ell\right), \text{LINEAR}\left(\boldsymbol{E}_c^\ell\right)\right), \text{ for } l = 1, 2, \ldots, L-1; \tag{6}$$

and a bottleneck block:

$$\boldsymbol{V} \leftarrow \text{MP} \circ \text{MP}\left(\boldsymbol{V} + \text{LINEAR}\left(\boldsymbol{V}_c^L\right), \text{LINEAR}\left(\boldsymbol{E}_c^L\right)\right). \tag{7}$$

The output features are passed through a node-wise MLP that returns $\boldsymbol{\mu}_i$ and $\boldsymbol{\sigma}_i$ for each node $i \in \mathcal{V}^L$. The latent variables are then computed as $\boldsymbol{\zeta}_i = \text{BATCHNORM}(\boldsymbol{\mu}_i + \boldsymbol{\sigma}_i \boldsymbol{\epsilon}_i)$, where $\boldsymbol{\epsilon}_i \sim \mathcal{N}(0, \boldsymbol{I})$. Finally, the decoder mirrors the encoder, employing a symmetric architecture (replacing graph pooling by graph unpooling layers) to upsample the latent features back to the original graph $\mathcal{G}$ (Figure 9c). Its blocks are also conditioned on the outputs of the condition encoder. The message passing and the graph pooling and unpooling layers in the VGAE are the same as in the (L)DGN.

The VGAE is trained to reconstruct states $\boldsymbol{Z}(t) \in \mathfrak{Z}$ with a KL-penalty towards a standard normal distribution on the learned latent space. Once trained, the LDGN can be trained following the approach in Section 3.1. However, the objective is now to learn the distribution of the latent states $\boldsymbol{\zeta}$, defined on the coarse graph $\mathcal{G}^L$, conditioned on the outputs $\boldsymbol{V}_c^L$ and $\boldsymbol{E}_c^L$ from the condition encoder. As illustrated in Figure 2b, during inference, the condition encoder generates the conditioning features $\boldsymbol{V}_c^\ell$ and $\boldsymbol{E}_c^\ell$ (for $l = 1, 2, \ldots, L$), and after the LDGN completes its denoising steps, the decoder transforms the generated $\boldsymbol{\zeta}_0$ back into the physical feature-space defined on $\mathcal{G}$.

Unlike in conventional VGAEs (Kipf & Welling, 2016), the condition encoder is necessary because, at inference time, we need an encoding of $\boldsymbol{V}_c$ and $\boldsymbol{E}_c$ on graph $\mathcal{G}^L$, where the LDGN operates. This

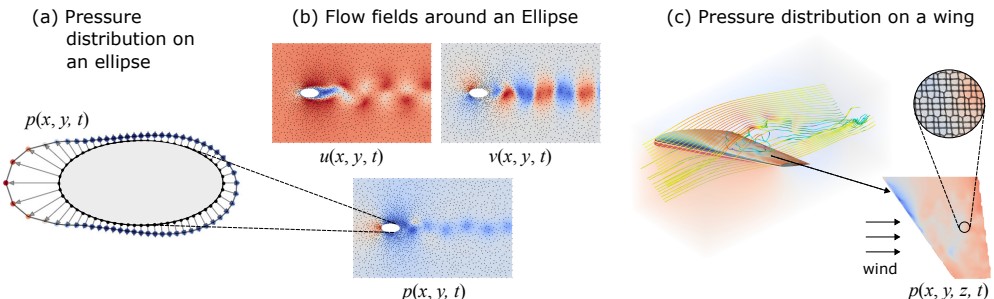

Figure 3: (L)DGNs can generate diverse states of dynamic mesh-based simulations given the mesh geometry and simulation parameters. We demonstrate this by learning: (a) the pressure on an ellipse in 2D laminar flow (ELLIPSE task), (b) the velocity and pressure fields resulting from that flow (ELLIPSEFLOW task), and (c) the pressure on a wing in 3D turbulent flow (WING task).

encoding cannot be directly generated by the encoder, as it also requires $\boldsymbol{Z}(t)$ as input, which is unavailable during inference. An alternative approach would be to define the conditions directly in the coarse representation of the system provided by $\mathcal{G}^L$, but this representation lacks fine-grained details, leading to sub-optimal results (see Appendix D.3 for details).

## 4  EXPERIMENTAL DOMAINS

We consider three experimental domains: (i) pressure on the wall of an ellipse in 2D quasi-periodic laminar flow (ELLIPSE task), (ii) velocity and pressure fields around that ellipse (ELLIPSEFLOW task), and (iii) pressure on a wing in 3D turbulent flow (WING task). The ELLIPSE and ELLIPSEFLOW data were adapted from Lino et al. (2022), while the dataset for the WING task was generated using Detached Eddy Simulation (DES) with OpenFOAM's PISO solver.

The ELLIPSEFLOW task involves a canonical fluid dynamics problem: predicting the velocity $\boldsymbol{u}$ and pressure $p$ fields around an elliptical cylinder (Figure 3b). While this task benefits from spatial refinement near the surface of the ellipse, the ELLIPSE task fully leverages the graph-based representation by focusing solely on the surface of the immersed object (Figure 3a). The WING experiments target wings in 3D turbulent flow, characterized by intricate vortices that spontaneously form and dissipate on the wing surface (Figure 3c). This task is particularly challenging due to the high-dimensional, chaotic nature of turbulence and its inherent multi-scale interactions across a wide range of scales. The geometry of the wings varies in terms of relative thickness, taper ratio, sweep angle, and twist angle. These simulations are computationally expensive, and using GNNs allows us to concentrate computational effort on the wing's surface, avoiding the need for costly volumetric fields. A regular grid around the wing would require over $10^5$ cells, in contrast to approximately 7,000 nodes for the surface mesh representation. The surface pressure can be used to determine both the aerodynamic performance of the wing and its structural requirements. Fast access to the probabilistic distribution of these quantities would be highly valuable for aerodynamic modeling tasks.

## 5  RESULTS

Our main findings indicate that DGNs and LDGNs can generate high-quality fields and accurately reproduce the distribution of converged states, even when trained on incomplete distributions. Both models outperformed the baselines – a vanilla GNN, a Bayesian GNN, a Gaussian regression GNN, a Gaussian mixture GNN, and a VGAE – in terms of sample accuracy and distributional accuracy. LDGNs showed improvement over DGNs, particularly in distributional accuracy and suppressing undesired high-frequency noise.

Unless otherwise specified, all models were trained on 10 consecutive states per system for the ELLIPSE and ELLIPSEFLOW domains, and 250 consecutive states (shortly after the data-generating simulator reached statistical equilibrium) for the WING domain. For the ELLIPSE and ELLIPSEFLOW domains, this covers approximately **26-48%** of the time points required to capture a full vortex-shedding cycle, while for the WING domain, it represents about **10%** of the time points needed to

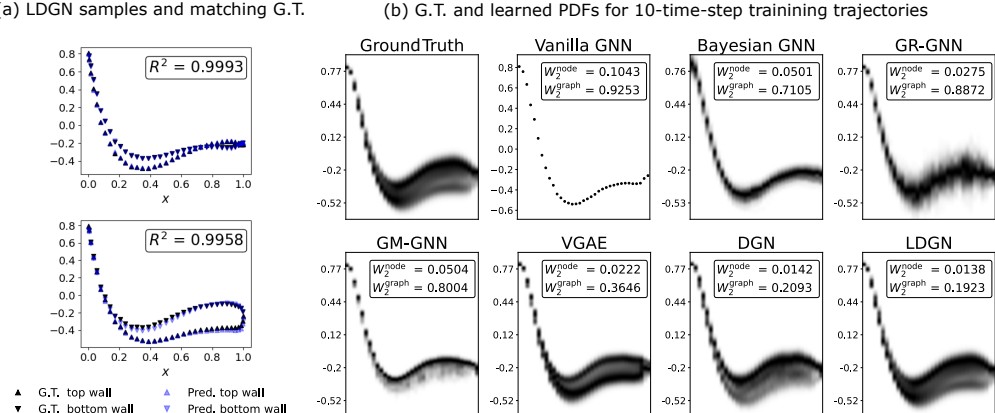

Figure 4: For a system on dataset ELLIPSE-INDIST, (a) samples from the LDGN and ground truth, and (b) probability density function from the DGN, LDGN, baseline models, and ground truth. The DGN and LDGN show the best distributional accuracy.

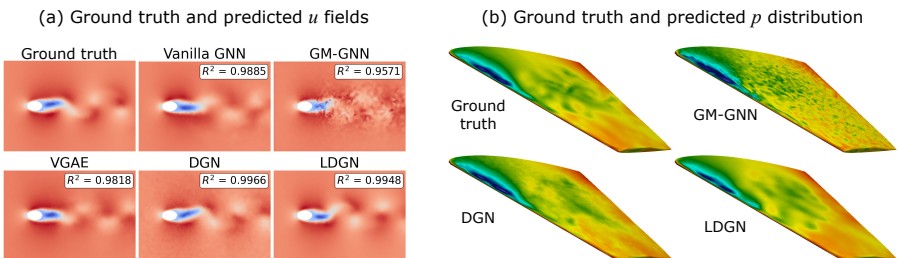

Figure 5: Samples from the DGN, LDGN, baseline models, and ground truth on (a) dataset ELLIPSEFLOW-INDIST, and (b) dataset WING-TEST. The DGNs and LDGNs achieve the highest sample accuracy, with the DGNs showing good accuracy but retaining some high-frequency noise.

achieve statistically stationary variance. To assess the performance, we compared to ground-truth trajectories spanning three complete periods and a time interval long enough to achieve stationary variance, respectively. To accelerate sampling, we adopted the strategy from Song & Ermon (2020), which considers only a subset of the denoising steps. We used 50 steps, evenly distributed.

**Sample accuracy**   Visually, the fields generated by our DGNs and LDGN models are faithful to the ground truth, as can be seen in Figures 4a and 5. In the ELLIPSE and ELLIPSEFLOW domains, it is possible to compare each sampled state with the states from a simulated trajectory and find its equivalent state (the one with the highest correlation), since these trajectories are smooth and periodic. For these domains, we report the coefficient of determination ($R^2$) between the generated samples and their ground-truth equivalents as a measure of the models' sample accuracy. In the ELLIPSE task, the LDGN achieved the highest accuracy, closely followed by the DGN (Table 5), and, in the ELLIPSEFLOW task, the DGN was the best performing model, followed by the LDGN (Table 6).

Although the DGN and LDGN models' accuracy is comparable, high-frequency noise is evident in the samples generated by the DGN model (Figure 5). To quantify high-frequency noise, we computed the graph Fourier transform of the predicted fields and assessed the energy associated with the 10 highest graph eigenvalues for all systems in the ELLIPSEFLOW-INDIST and WING-TEST datasets. We found that the DGNs consistently over-estimated high-frequency content in generated samples, while LDGN samples remained close to the ground truth, with on average 3x reduced high-frequency error (See Appendix D.4 for detailed numbers). This discrepancy is due to the fact that DGNs are required to handle high-frequency details directly, whereas LDGNs leverage a VGAE to transform potentially noisy latent features back to physical space. The VGAE, being robust to noisy latent features, effectively mitigates such noise.

**Distributional accuracy** High sample accuracy does not necessarily imply that a model is learning the true distribution. In fact, these properties often conflict. For instance, in VGAEs, the KL-divergence penalty allows control over whether to prioritize sample quality or mode coverage. To evaluate how well models capture the probability distribution of system states, we use the Wasserstein-2 distance. This metric can be computed in two ways: (i) by treating the distribution at each node independently and averaging the result across all nodes, or (ii) by considering the joint distribution across all nodes in the graph. We denote these metrics as $W_2^{\text{node}}$ and $W_2^{\text{graph}}$, respectively. The node-level measure ($W_2^{\text{node}}$) provides insights into how accurately the model estimates point-wise statistics, such as the mean and standard deviation at each node. However, it does not penalize inaccurate spatial correlations, whereas the graph-wise measure ($W_2^{\text{graph}}$) does. To ensure stable results when computing these metrics, the target distribution for the ELLIPSE and ELLIPSE-FLOW tasks is represented by 60 to 100 consecutive states, depending on the period length, while the predicted distribution is represented by 200 samples. For the WING task, the target distribution is represented by 2,500 consecutive states, and the predicted one by 3,000 samples.

In our experiments, the training trajectories are intentionally short. For instance, in the ELLIPSE-TRAIN and ELLIPSEFLOW-TRAIN datasets, the trajectories are too short to cover one full oscillation period, meaning they do not explicitly provide full statistical information about the systems. In the WING-TRAIN dataset, while the trajectories are long enough to capture the mean flow, they fall short of capturing the standard deviation, spatial correlations, or higher-order statistics. Despite these challenges, the DGN, and especially the LDGN, are capable of accurately learning the complete probability distributions of the training trajectories and accurately generating new distribution

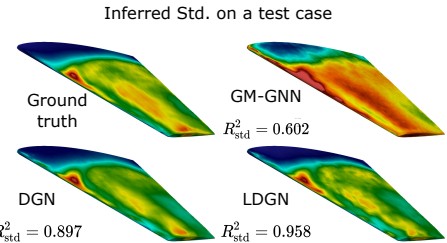

Figure 6: The LDGN's standard deviation is the closest to the ground-truth.

for both in- and out-of-distribution physical settings. This capability is highlighted in Figure 7a, which shows the probability density function (PDF) for a training trajectory of the ELLIPSE-TRAIN dataset (top right) and the PDF learned by the LDGN for the same system (bottom right). Similarly, Figure 1b demonstrates how the LDGN accurately captures the standard deviation across all possible states (bottom right), even though this information is not provided in the training dataset. This excellent distributional accuracy was likewise observed for test data, Figures 4b and 6. As demonstrated in Figure 4b, this is a rare property – (L)DGN is the only method which could faithfully represent the ground-truth distribution, while other methods show either collapse or wrong weighting.

The Wasserstein-2 distances for all datasets are summarized in Tables 7 to 9. The LDGN consistently outperforms other models, followed by the DGN. Experiments in Appendix D.5 suggest that the ability of the diffusion-based models to learn the full distribution from short trajectories likely stems from their capacity to recognize common physical patterns across different systems, leveraging this knowledge to enhance its understanding of each individual trajectory. Unlike models that rely on sampling from a compressed latent space or make assumptions about probability distributions for parameters or outputs, their gradual denoising approach minimizes overfitting based on only the training data. This allows them to extrapolate the training trajectories and generalize to new physical settings more effectively. The LDGN variant outperforms the DGN thanks to learning the distributions in a more meaningful latent space that is less sensitive to small-scale fluctuations.

**Generalization** We tested generalization to out-of-distribution Reynolds numbers, ellipse thickness and ellipses rotated by 10-degrees in the ELLIPSE and ELLIPSEFLOW tasks. The DGN and LDGN models generally exhibit good generalization, as shown by their sample and distributional accuracy in the out-of-distribution datasets, which is comparable to their performance on the in-distribution datasets (Tables 5 to 8). However, we observed that both types of accuracy degrade as the variance of the ground-truth trajectories increases, even within the in-distribution datasets, as seen in Figure 12. This is understandable, as higher diversity of states makes it more difficult for the model to identify common features among systems that are close in parameter space, thereby affecting sample quality. Additionally, low-probability states in high-variance trajectories are rarely encountered during training, and can differ significantly from those in lower-variance trajectories, such as states of separated flow near the ellipse's leading edge. As a result, these states may be overlooked by the models, leading to reduced distributional accuracy.

**Influence of the number of states per training trajectory**   A key strength of the DGN and LDGN models is their ability to learn all possible states of a system from incomplete data. In the ELLIPSE task, we used 10 consecutive time-steps per training trajectory, while the full periods of these trajectories range from 21 to 38 time-steps. We found that the distributional accuracy of the DGN and LDGN models is only slightly affected by further reducing the length of the training trajectories, and, interestingly, this accuracy remains relatively close to the accuracy of training runs with complete periods. This behavior is illustrated in Figure 7b, which shows the $W_2^{\text{node}}$ and $W_2^{\text{graph}}$ metrics for the DGN and LDGN models, as well as for four baseline models, on the ELLIPSE-INDIST dataset. In this figure, the darker gray region indicates training trajectories that are always longer than the oscillation period, while in the lighter gray region at least one trajectory exists that is long enough. To the left of this area, it can be observed that the $W_2^{\text{node}}$ of most baselines drops sharply due to the missing information. In contrast, the DGN and LDGN models are barely affected. It is also worth noting that the Gaussian mixture model (GM-GNN) achieves the lowest $W_2^{\text{node}}$ among all models when trained on long trajectories. However, this baseline does not model spatial correlations, which explains its consistently high $W_2^{\text{graph}}$ values.

**Computational efficiency**   Direct comparisons between very different implementations are inherently difficult. Nonetheless, to provide rough estimates of the performance characteristics for our WING experimental domain, the ground-truth simulator, running on 8 CPU threads, required 2,989 minutes to simulate the initial transient phase plus 2,500 equilibrium states – sufficient to obtain a converged variance. In contrast, the LDGN model took only 49 minutes on 8 CPU threads and 2.43 minutes on a single GPU to generate 3,000 samples. If we consider the generation of a single converged state (for use as an initial condition in another simulator, for example), the speedup is four orders of magnitude on the CPU and five orders of magnitude on the GPU. Thanks to its latent space, the LDGN model is not only more accurate, but also $8\times$ faster than the DGN model, while requiring only about 55% more training time. Performance details are available in Appendix D.1. This significant efficiency advantage suggests that these models could be particularly valuable in scenarios where computational costs are otherwise prohibitive.

**Multi-scale vs. single-scale DGN**   A key distinction of our GNN architecture is the use of a multi-scale GNN, while all previous works applied message passing directly on the input graph (Hoogeboom et al., 2022; Wu et al., 2024; Yi et al., 2024; Wen et al., 2023). DDPMs are known to benefit from hierarchical and global denoising transitions, which enable the model to capture and process information at multiple spatial resolutions (Dhariwal & Nichol, 2021; Si et al., 2023). While the previously used single-scale GNNs may suffice for smaller systems, we observed that they failed on larger systems, such as those in the ELLIPSEFLOW and WING domains, due to the limited reach of their receptive field. For a fair comparison to heterogeneous previous work, we have considered a single-scale version of our DGN (with the same parametrization and hyperparameters). In the ELLIPSE task (approximately 70 nodes per graph), our proposed four-scale DGN significantly outperformed its single-scale counterpart in both sample and distributional accuracy, as shown in

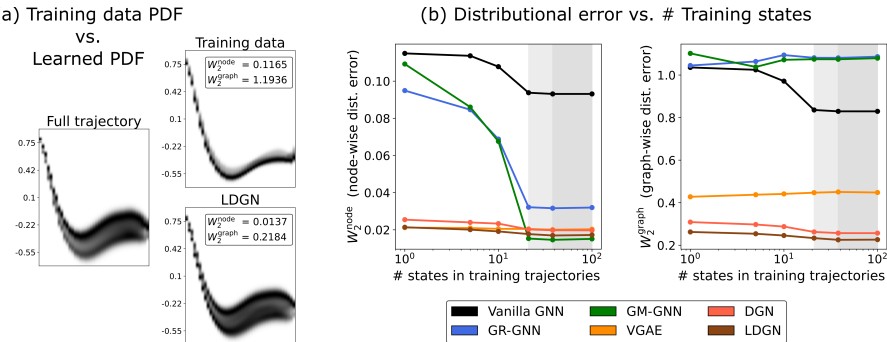

Figure 7: (a) The LDGN accurately learns the full probability distribution of the ELLIPSE system states, despite being trained on short trajectories (10 consecutive snapshots) that do not cover all possible states. (b) The distributional error of the DGN and LDGN models is not significantly affected by short training trajectories. They consistently exhibit the lowest $W_2^{\text{graph}}$ distance.

Table 11 of the appendix. In the ELLIPSEFLOW task ($\sim$2.3k nodes per graph) and the WING task ($\sim$6.8k nodes), the single-scale version did not produce feasible solutions (see Figure 15). We observed that, in the ELLIPSEFLOW task, performance improves as the number of scales is increased , with the five-scale model yielding the best results, as also shown in Table 11. This model is also the most computationally efficient, outperforming the two-scale version by 25%.

**Comparison to probabilistic baselines**   We compare to a Bayesian GNN, a Gaussian mixture model (GM-GNN), a Gaussian regression model (GR-GNN), and a VAE on the ELLIPSE task; to a GM-GNN and a VAE on the ELLIPSEFLOW task; and to a GM-GNN on the WING task.

*Bayesian GNN:*  Although the Bayesian GNN provided reasonable predictions for both in- and out-of-distribution datasets, and achieved higher sample and distributional accuracy than other baselines, its performance was still substantially inferior to that of the DGN and LDGN models (see Tables 5 and 7). Additionally, the probability distribution of the outputs incorrectly resembles a Gaussian distribution (Figure 5). Another significant drawback is its high training cost (approximately 8 times slower than the DGN training), making it impractical for larger physical systems.

*Gaussian Mixture Model:*  The GM-GNN represents a learned distribution as a combination of multiple Gaussian distributions. For each node it provides the mean, variance, and weight characterizing each Gaussian distribution. Once the trained GNN is evaluated for a given conditional input, sampling is efficient. However, since it is performed independently for each node, the resulting samples are spatially discontinuous (Figure 5) and fail to represent the underlying distribution (Figure 4b). This leads to low sample accuracy (Tables 5 and 6) and a high $W_2^{\text{graph}}$ (Tables 7 to 9). Interestingly, in the ELLIPSE task, the GM-GNN outperforms the DGN and LDGN models in terms of $W_2^{\text{node}}$, a metric that does not penalize discontinuous samples, when trained on complete trajectories (grey region in Figure 7b). However, when trained on shorter trajectories, its performance clearly deteriorates (white region in Figure 7b). This suggests that the GM-GNN is less capable at learning the shared physical features across different systems. In comparison, the DGN and LDGN models are much better suited for learning from partial distributions, which is especially relevant for 3D chaotic systems due to the high cost of data generation. We also considered the GR-GNN as a special case of the GM-GNN. It represents the learned distribution as a single Gaussian. However, it generally exhibited a worse performance than the GM-GNN.

*Variational Autoencoder:*  Finally, a VGAE was also considered as a baseline model. This model mirrors the architecture of the VGAE proposed for the LDGN (Section 3.2). but with the number of message-passing and graph-pooling layers matching the totals used in the (L)DGN models. This baseline achieved a sample accuracy closer to that of the DGN and LDGN models (Tables 5 and 6). In the ELLIPSE task, it successfully learned the distribution of states when trained on short trajectories (Figures 4b and 7b). However, in the more complex ELLIPSEFLOW task, it suffered from mode collapse despite careful selection of the latent space size and the KL penalty (Table 8). We believe that training on short trajectories caused the decoder to become overly dependent on the input conditions rather than on the latent features. In contrast, the VGAE in the LDGN models did not encounter this issue, as their latent space is less compressed.

## 6   CONCLUSIONS

We have introduced physical- and latent-space graph-based diffusion models that enable direct sampling of physical states in large, geometrically complex dynamical systems from their equilibrium distribution. This approach allows for the efficient computation of flow statistics without the need for long and expensive numerical simulations. Our latent-space model produces high-quality solutions without undesired high-frequency noise in a fraction of the time compared to the physical-space model. Notably, it accurately learns full distributions even when trained on incomplete data from short simulations, marking an important step towards more efficient probabilistic modeling. However, our method has limitations that suggest interesting directions for future work. For example, the samples produced by our method are not temporally correlated, and it would be valuable to extend the model with temporal conditioning. Additionally, incorporating physical constraints (e.g., through guidance) could further improve the quality of the generated solutions (Huang et al., 2024; Lino et al., 2024). Another promising direction is improving sampling speed, which could potentially be achieved through flow matching (Lipman et al., 2023) – see Appendix D.7. Despite these challenges, we believe our work represents a significant step towards leveraging diffusion models in real-world engineering applications.

## ACKNOWLEDGMENTS

M.L. and N.T. acknowledge the support of the European Research Council (ERC) Consolidator Grant SpaTe (No. CoG-2019-863850). The authors gratefully acknowledge the Gauss Centre for Supercomputing e.V. (www.gauss-centre.eu) for providing computing time on the GCS Supercomputer SuperMUC at the Leibniz Supercomputing Centre (www.lrz.de). M.L. also expresses gratitude to Qiang Liu for his valuable advice and discussions on diffusion models.

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

## A  DENOISING DIFFUSION PROBABILISTIC MODELS (DDPMS)

DDPMs can learn to synthesise a sample $z^0$ from a data distribution $q(z^0)$ by progressively denoising a sample $z^R$ drawn from an isotropic Gaussian distribution (Song & Ermon, 2020; Ho et al., 2020). This denoising process mirrors the inverse of a fixed Markov chain of length $R$, known as the *forward process*, and it consists of $R$ denoising steps. The forward (or *diffusion*) process produces variables $z^1$ through $z^R$ by sequentially injecting them with Gaussian noise:

$$q(z^r|z^{r-1}) := \mathcal{N}(z^r; \sqrt{1-\beta_t}z^{r-1}, \beta_r\mathbf{I}), \tag{8}$$

where $\beta_r \in (0,1)$. Thanks to the reparametrization trick (Kingma et al., 2015), it is possible to shortcut in the forward process and directly sample $z^r$, at any diffusion step $r$, according to

$$q(z^r|z^0) = \mathcal{N}(z^r; \sqrt{\bar{\alpha}_r}z^0, (1-\bar{\alpha}_r)\mathbf{I}), \tag{9}$$

this is,

$$z^r = \sqrt{\bar{\alpha}_r}z^0 + \sqrt{1-\bar{\alpha}_r}\epsilon. \tag{10}$$

Here, $\alpha_r := 1 - \beta_r$, $\bar{\alpha}_r := \prod_{s=1}^r \alpha_s$ and $\epsilon \sim \mathcal{N}(\mathbf{0}, \mathbf{I})$ (Ho et al., 2020). If $R$ is large enough and the $\beta$-schedule is selected properly, the forward process leads to $z^R$ (nearly) following a Gaussian distribution. Typical values for $R$ are around thousands, and linear and cosine $\beta$-schedules are the most common (Nichol & Dhariwal, 2021).

Given the data distribution $q(z^0)$, the transitions of the reverse (or *denoising*) process $q(z^{r-1}|z^r)$ can be approximated using a neural network parametrised by $\theta$,

$$p_\theta(z^{r-1}|z^r) := \mathcal{N}(z^r; \mu_\theta(z^r, r), \Sigma_\theta(z^r, r)). \tag{11}$$

Although the mean $\mu_\theta(z^r, r)$ could be parametrised in several ways, Ho et al. (2020) found best to first predict the noise $\epsilon$ with the neural network and then compute $\mu_\theta(z^r, r)$ according to

$$\mu_\theta(z^r, r) = \frac{1}{\sqrt{\alpha_r}}\left(z^r - \frac{\beta_r}{\sqrt{1-\bar{\alpha}_r}}\epsilon_\theta(z^r, r)\right). \tag{12}$$

To simplify training, Ho et al. (2020) avoided learning the variance and fixed it to $\Sigma_\theta(z^r, r) = \sigma_r^2\mathbf{I}$, with either

$$\sigma_r^2 = \beta \quad \text{or} \quad \sigma_r^2 = \tilde{\beta}_r = (1-\bar{\alpha}_{r-1})/(1-\bar{\alpha}_r)\beta_r,$$

which correspond to the upper and lower bounds on the reverse process entropy, respectively. Following Nichol & Dhariwal (2021), we opt to parameterise $\Sigma_\theta(z^r, r)$ as an interpolation between the lower, $\tilde{\beta}_r$, and upper, $\beta_r$, bounds on the reverse process entropy in the log domain:

$$\Sigma_\theta(z^r, r) = e^{\mathbf{v}_\theta(z^r, r)\log\beta_r + (1-\mathbf{v}_\theta(z^r, r))\log\tilde{\beta}_r}, \tag{13}$$

where $\mathbf{v}_\theta(z^r, r)$ is predicted by the neural network. Nichol & Dhariwal (2021) demonstrated that learning $\Sigma_\theta(z^r, r)$ improved the likelihood of DDPMs and reduced the number of diffusion steps required for sampling from the learned $q(z^0)$ distribution at inference, which we considered especially relevant for flow field synthesis due to the time constraints in design optimisation tasks.

To train a network to approximate $\epsilon_\theta(z^r, r)$ and $\mathbf{v}_\theta(z^r, r)$, we minimise the loss function $\mathcal{L}$, defined as the sum of two terms:

$$\mathcal{L} = \mathcal{L}_{\text{simple}} + \lambda_{\text{vlb}}\mathcal{L}_{\text{vlb}}, \tag{14}$$

where $\mathcal{L}_{\text{simple}}$ and $\mathcal{L}_{\text{vlb}}$ are defined as

$$\mathcal{L}_{\text{simple}} = E_{r,z^0,\epsilon}\left[||\epsilon - \epsilon_\theta(z^r, r)||^2\right], \tag{15}$$

$$\mathcal{L}_{\text{vlb}} = -\log p_\theta(z^0|z^1) + \sum_2^T D_{KL}\left(q(z^{t-1}|z^t, z^0) \,||\, p_\theta(z^{t-1}|z^t)\right).$$

The term $\mathcal{L}_{\text{vlb}}$ represents the variational lower bound. Its first term is the negative log-likelihood of a Gaussian distribution, while the remaining terms are the Kullback-Leibler (KL) divergence between two Gaussians. The gradients of $\epsilon_\theta(z^r, r)$ are only backpropagated through $\mathcal{L}_{\text{simple}}$, whereas the $\mathcal{L}_{\text{vlb}}$ term is used to optimise $\mathbf{v}_\theta(z^r, r)$. This training strategy was proposed by Nichol & Dhariwal (2021) and has also been proven successful in subsequent work Dhariwal & Nichol (2021).

To reduce gradient noise, we employ importance sampling and dynamically weight each loss term based on the loss history (Nichol & Dhariwal, 2021).

## B ADDITIONAL MODEL DETAILS

The implementation of our models and baselines, including their weights, and demonstration scripts are available at https://github.com/tum-pbs/dgn4cfd.

### B.1 (L)DGN ARCHITECTURE AND TRAINING

The message-passing layers in the (L)DGN's processor follow the general framework described in Battaglia et al. (2016) and Battaglia et al. (2018). The edge- and node-update functions are modeled by MLPs with one hidden layer, containing $F_h$ neurons, and with $F_h$ output features. Their activation functions are SELU functions with their standard parameters (Klambauer et al., 2017) – as are all the activations in the model. These MLPs are preceded by layer normalization (Ba et al., 2016). Specifically, the edge-update, edge-aggregation, and node-update steps are as follows:

$$e_{ij} \leftarrow W_e e_{ij} + \mathrm{MLP}^e \left( \mathrm{LN} \left( [e_{ij}|v_i|v_j] \right) \right), \qquad \forall (i,j) \in \mathcal{E}, \tag{16}$$

$$\bar{e}_j \leftarrow \sum_{i \in \mathcal{N}_j^-} e_{ij}, \qquad \forall j \in \mathcal{V}, \tag{17}$$

$$v_j \leftarrow W_v v_j + \mathrm{MLP}^v \left( \mathrm{LN} \left( [\bar{e}_j|v_j] \right) \right), \qquad \forall j \in \mathcal{V}, \tag{18}$$

The (L)DGN processes information at $L$ length scales by creating lower-resolution graphs and propagating the node and edge features across them. These low-resolution graphs have fewer nodes and edges than the original mesh graph, allowing a single message-passing layer to propagate attributes over longer distances more efficiently. In the (L)DGN's processor, information is first distributed and processed in the high-resolution graph through two sequential message-passing layers. It is then passed to the immediately lower-resolution graph via graph pooling (equations (21) and (20)), and, here, the features are again processed through two message-passing layers. This process is repeated $L - 1$ times in total. Once the lowest-resolution features are processed at $\mathcal{G}^L$, they are passed back to the scale immediately above through a graph-unpooling layer (equation (22)), which also takes as input the node-feature values at this scale. The features are successively passed through pairs of message-passing layers and unpooling layers until the information is processed again at the original mesh graph.

For a fair comparison all DGN, LDGN and baselines models in Appendices B.5, D.2 and D.3, trained for the same task, follow a similar architecture, with the same number of message-passing layers and a comparable number of weights. In the ELLIPSE task, all models operate across 4 levels of resolution (14 message-passing layer in total) and have approximately 3.5M parameters. In the ELLIPSEFLOW task, they span 5 levels of resolution (18 message-passing layer in total) and have 4.5M parameters, while in the WING task, they cover 6 levels of resolution (22 message-passing layer in total) with 5.5M parameters. In the LDGN models, the VGAE applies two graph-pooling layers before denoising (Figure 8), and the denoising model applies the remaining graph-pooling layers until the target resolution is achieved. In the ELLIPSE task (1D meshes), this VGAE provides approximately 4-fold compression, whereas in the ELLIPSEFLOW and WING tasks (2D meshes), it roughly offers a 16-fold compression.

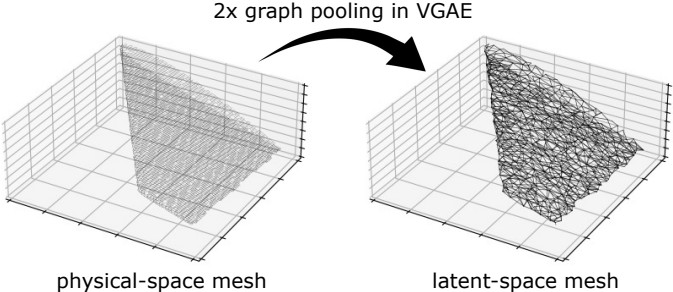

Figure 8: In our experiments, the LDGN models are applied to the latent space of two-scale VGAEs.

The DGN and LDGN models were trained using the node-wise mean of the loss function described in equation (14). To reduce gradient noise and improve training stability, we employed the impor-

tance sampling technique from Nichol & Dhariwal (2021). During each training epoch, a single state was sampled from each training trajectory.

The initial learning rate was set to $10^{-4}$ and was reduced by a factor of 10 when the training loss plateaued for $n$ consecutive epochs. In the ELLIPSE and ELLIPSEFLOW tasks, we used $n = 50$, while in the WING task, $n$ was set to 250 due to the shorter length of the training dataset. Training continued until the learning rate reached $10^{-8}$. This training strategy was consistently applied to both the DGN, LDGN, and baseline models.

## B.2    VGAE ARCHITECTURE AND TRAINING

The LDGN's VGAE consists of a condition encoder, a node encoder, and a node decoder, as illustrated in Figure 9 and outlined in Section 3.2. It was trained to reconstruct the input node features with a low-weighted KL term between the latent distribution of each node, $\zeta_i$, and a standard normal distribution:

$$\mathcal{L}_{\text{VGAE}} = \frac{1}{|\mathcal{V}|} \sum_{i \in \mathcal{V}} ||z_i - z_i'||^2 + 10^{-6} \times \left( -\frac{1}{2|\mathcal{V}|} \sum_{i \in \mathcal{V}} \left( 1 + \log\left(\sigma_i^2\right) - \mu_i^2 - \sigma_i^2 \right) \right) \quad (19)$$

Figure 9: Diagram of the VGAE architecture. It consist of (a) a condition encoder, (b) a node encoder, and (c) a node decoder.

## B.3    GRAPH POOLING AND UNPOOLING

Our DGN, LDGN, and baseline models (except for the VGAE, since there are no skip connections between the encoder and decoder branches) follow a U-Net-like architecture (Ronneberger et al., 2015), where message passing is applied across $L$ levels of resolution (Gao & Ji, 2019; Lino et al., 2022; Fortunato et al., 2022; Cao et al., 2023). Thus, in addition to the primal graph, $G := (\mathcal{V}, \mathcal{E})$, $L - 1$ lower-resolution graphs are employed. Here, for each resolution level $\ell$ from 2 to $L$, we denote their graphs as $\mathcal{G}^\ell := (\mathcal{V}^\ell, \mathcal{E}^\ell)$, while the primal graph is denoted as $\mathcal{G} \equiv \mathcal{G}^1 := (\mathcal{V}^1, \mathcal{E}^1)$. To generate each lower-resolution graph $G^\ell$, we apply Guillard's coarsening algorithm (Guillard, 1993), originally developed for multi-grid methods in CFD. This algorithm removes a subset of nodes from the input mesh (or bi-directional graph) as outlined in Algorithm 1. For 1D meshes, the compression ratio is approximately two, and for 2D meshes, around four.

Unlike the methods in Gao & Ji (2019) and Lino et al. (2022), our coarsening approach preserves the original relative distribution of nodes in the coarser graphs. Furthermore, for the WING task, the voxel-grid coarsening used in Lino et al. (2022) introduces new nodes and edges that do not lie on the WING surface. In contrast, Guillard's coarsening ensures that the coarser nodes and edges remain strictly on the Wing surface, thereby avoiding undesired connections between nodes that may be spatially close in the world-space but distant in the mesh-space (e.g., a pair of nodes located on the upper and lower surfaces of the Wing). Additionally, unlike the method in Fortunato et al. (2022), our approach operates automatically without relying on an external mesh generator. Another valid and similar alternative could have been bi-stride pooling (Cao et al., 2023).

In our implementation, once $\mathcal{V}^{\ell+1} \subset \mathcal{V}^\ell$ is obtained from $\mathcal{G}^\ell$, each node $i \in \mathcal{V}^\ell$ is assigned a *parent* node $j \in \mathcal{V}^{\ell+1}$, denoted as $\mathcal{P}_i$. Each node $i$ is assigned to the parent node that is the fewest

---

**Algorithm 1** Guillard's coarsening algorithm (Guillard, 1993)

---

1: mask ← ones($|\mathcal{V}^\ell|$)                                                              ▷ Vector of size $|\mathcal{V}^\ell|$ filled with ones
2: **for** node $i \in \mathcal{V}^\ell$ **do**                                                    ▷ Iterate node-by-node
3:     **if** mask$[i] = 1$ **then**                                       ▷ If first visit to node $i$ then this node is not dropped
4:         **for** node $j \in \mathcal{N}_i^-$ **do**
5:             mask$[j] \leftarrow 0$                                                   ▷ The incoming neighbours are dropped
6:         **end for**
7:     **end if**
8: **end for**
9: $\mathcal{V}^{\ell+1} \leftarrow \mathcal{V}^\ell[$mask$]$

---

hops away. If multiple candidates are equidistant, the parent node is chosen based on the smallest Euclidean distance. Nodes that share the same parent node $j \in \mathcal{V}^{\ell+1}$ are referred to as its *child* nodes, denoted by $\mathcal{C}h_j \subset \mathcal{V}^\ell$. The set of edges $\mathcal{E}^{\ell+1}$ is constructed to preserve the connectivity of the child nodes in $\mathcal{V}^\ell$. Specifically, there is an edge from $i \in \mathcal{V}^{\ell+1}$ to $j \in \mathcal{V}^{\ell+1}$ if at least one edge exists between their child nodes in $\mathcal{V}^\ell$. All these operations are performed as pre-processing steps and do not occur during model evaluation.

To implement a U-Net-like architecture, we use graph pooling and unpooling to transition between higher and lower-resolution graphs. Following Lino et al. (2022) and Fortunato et al. (2022), both pooling and unpooling operations are based on message passing.

**Graph pooling**     Graph pooling from $\mathcal{V}^\ell$ to $\mathcal{V}^{\ell+1}$ is performed by message passing from each node $i \in \mathcal{C}h_j$ to its parent node $j \in \mathcal{V}^{\ell+1}$, following equation (20). Edge features $\mathcal{E}^{\ell+1}$ are assigned via linear projection of the relative positions between parent nodes, as shown in equation (21).

$$\boldsymbol{v}_j \leftarrow \sum_{i \in \mathcal{C}h_j} \text{MLP}\left(\text{LN}\left([\text{Linear}(\boldsymbol{x}_j - \boldsymbol{x}_i)|\boldsymbol{v}_i]\right)\right), \qquad \forall j \in \mathcal{V}^{\ell+1}, \tag{20}$$

$$\boldsymbol{e}_{jk} \leftarrow \text{Linear}(\boldsymbol{x}_j - \boldsymbol{x}_k), \qquad \forall (j,k) \in \mathcal{E}^{\ell+1}. \tag{21}$$

**Graph unpooling**     Similarly, graph unpooling from $\mathcal{V}^\ell$ to $\mathcal{V}^{\ell-1}$ is performed by message passing to each node $i \in \mathcal{V}^{\ell-1}$ from its parent node $\mathcal{P}_i \subset \mathcal{V}^\ell$, as defined in equation (22):

$$\boldsymbol{v}_i \leftarrow \text{MLP}\left(\text{LN}\left([\text{Linear}(\boldsymbol{x}_i - \boldsymbol{x}_j)|\boldsymbol{v}_j|\boldsymbol{v}_i]\right)\right), \qquad \forall i \in \mathcal{V}^{\ell-1} \quad \text{and} \quad j = \mathcal{P}_i. \tag{22}$$

The features of $\mathcal{E}^{\ell-1}$ come via a skip connection from the encoder branch, just before the graph-pooling layer from $\mathcal{V}^{\ell-1}$ to $\mathcal{V}^\ell$ is applied.

### B.4    TREATMENT OF DIRICHLET BOUNDARY CONDITIONS

In the ELLIPSEFLOW task, the velocity values at the inlet and on the ellipse wall are known a priori, defining Dirichlet boundary conditions. To ensure that these boundary conditions are exactly satisfied in the DGN's samples, we apply deterministic diffusion transitions to the nodes on these boundaries. Specifically, for boundary nodes, we propagate the values deterministically as $\boldsymbol{v}_i^r := \sqrt{\alpha_r}\,\boldsymbol{v}_i^{r-1}$, and instead of learning their denoising transitions, we directly reverse the diffusion process, using $\boldsymbol{v}_i^{r-1} := \boldsymbol{v}_i^r/\sqrt{\alpha_r}$.

During training, the loss function is not evaluated at these boundary nodes, and during inference, their noisy node features are sampled not from a Gaussian distribution but are instead assigned their "diffused" values, i.e., $\boldsymbol{v}_i^R := \sqrt{\bar{\alpha}_R}\,\boldsymbol{v}_i^0$. While more sophisticated methods exist for handling boundary conditions, such as those proposed by Lugmayr et al. (2022), our approach remains efficient and straightforward.

In the LDGN and baseline models, we directly replace the output at these boundary nodes with their right values.

Table 1: Model size and hyper-parameters

| Task | Model | # Parameters | $F_h$ | $F_{\text{emb}}$ | $F_{\text{ae}}$ | $F_L$ | # Scales |
|---|---|---|---|---|---|---|---|
| ELLIPSE | DGN | 3.51 M | 128 | 512 | – | – | 4 |
| | LDGN | 3.52 M | 128 | 512 | 126 | 1 | 2 + 2 |
| | Vanilla GNN | 3.53 M | 154 | – | – | – | 4 |
| | Bayesian GNN | 3.56 M | 138 | – | – | – | 4 |
| | GR-GNN | 3.52 M | 154 | – | – | – | 4 |
| | GM-GNN | 3.53 M | 154 | – | – | – | 4 |
| | VGAE | 3.48 M | 124 | – | – | 4 | 4 |
| ELLIPSEFLOW | DGN | 4.50 M | 128 | 512 | – | – | 5 |
| | LDGN | 4.51 M | 128 | 512 | 126 | 1 | 2 + 3 |
| | Vanilla GNN | 4.50 M | 153 | – | – | – | 5 |
| | GM-GNN | 4.50 M | 153 | – | – | – | 5 |
| | VGAE | 4.51 M | 126 | – | – | 32 | 5 |
| WING | DGN | 5.48 M | 128 | 512 | – | – | 6 |
| | LDGN | 5.49 M | 128 | 512 | 126 | 1 | 2 + 4 |
| | GM-GNN | 5.45 M | 152 | – | – | – | 6 |

## B.5 BASELINE DETAILS

We compared the DGN and LDGN models with several GNN-based probabilistic baseline models: a Vanilla GNN, a Bayesian GNN, a Gaussian Regression GNN, a Gaussian Mixture GNN, and a VGAE. Details on these baselines are provided below.

**Vanilla GNN**   The so-called Vanilla GNN is a deterministic GNN that mirrors the architecture of the DGN, except for the absence of the diffusion-step encoder and the $r_{\text{emb}}$ conditioning layers. It takes the conditioning features $V_c$ and $E_c$ as input and returns constant (for the given conditions) node features, $z_i'$, which represent the mean learned from the training data. It was trained to maximize the expected data, this is

$$\mathcal{L}_{\text{vanilla}} = \frac{1}{|\mathcal{V}|} \sum_{i \in \mathcal{V}} ||z_i - z_i'||^2, \tag{23}$$

where $z_i$ are the states sampled from the training trajectories. Vanilla GNNs were trained for the ELLIPSE and ELLIPSEFLOW tasks. Since it was trained on short trajectories (10 snapshots per trajectory), its predictions resembles a single state rather than the mean of the complete distribution. This explains the relatively high sample accuracy of these models, while their distributional accuracy is, of course, low (Tables 5 to 8).

**Bayesian GNN**   In Bayesian neural networks, prediction variability arises from modeling the parameters as random variables with associated probability distributions. We compared our DGN and LDGN models to a Bayesian GNN on the ELLIPSE task. To train the Bayesian GNN efficiently, we used variational inference and approximated the posterior distribution of the parameters with a Gaussian. The architecture of the Bayesian GNN mirrors that of the DGN, except for the absence of the diffusion-step encoder and the $r_{\text{emb}}$ conditioning layers. During training, we maximized the Evidence Lower Bound (ELBO), assuming a Gaussian prior distribution over the parameters. We performed a grid search to find the optimal weight for the KL-divergence term in the ELBO loss (Liu & Thuerey, 2024), which was determined to be 0.1 for maximizing distributional accuracy on the ELLIPSE-INDIST dataset.

**Gaussian Mixture GNN (GM-GNN)**   A Gaussian mixture model (GMM) represents a learned distribution as a combination of multiple Gaussian distributions (Maulik et al., 2020). Its architecture mirrors that of the DGN. Through grid search, we selected three Gaussian components for the

ELLIPSE and ELLIPSEFLOW tasks, and five components for the WING task. Once the GNN is evaluated for a given conditional input, sampling is efficient: a Gaussian distribution is selected based on the returned weights, followed by sampling from that distribution.

**Gaussian Regression GNN (GR-GNN)**   We also considered a Gaussian Regression model, referred to as GR-GNN, which is a special case of the GM-GNN that represents the learned distribution as a single Gaussian.

**VGAE**   As a VGAE baseline, we considered a model mirroring the architecture of the VGAE proposed for the LDGN, but with the number of message-passing and graph-pooling layers matching the total used in the DGN models. For the ELLIPSE task, we set the size of the latent features to $F_L = 1$ and performed a grid search to find the optimal weight for the KL term (Kingma et al., 2015), which was determined to be 0.001 to maximize distributional accuracy on the ELLIPSE-INDIST dataset. For the ELLIPSEFLOW task, we found the best KL weight to be $10^{-8}$, and the optimal size of the latent features is $F_L = 32$ to maximize distributional accuracy on the ELLIPSEFLOW-INDIST dataset.

The number of hidden features per node and edge after encoding is denoted as $F_h$, and the size of the diffusion-step embedding is represented by $F_{\text{emb}}$. In the LDGNs' VGAE, the intermediate hidden features are denoted by $F_{\text{ae}}$, and the number of latent features by $F_L$. The values of these hyperparameters for each model, as well as the total model size, are summarized in Table 1.

## C   EXPERIMENTAL AND DATASET DETAILS

The datasets used in our ELLIPSEFLOW and ELLIPSE experiments are available at https://huggingface.co/datasets/mariolinov/Ellipse, and the datasets used in our WING experiments are available at https://huggingface.co/datasets/mariolinov/Wing. To the best of our knowledge, no previous studies have tackled large-scale, unsteady, and turbulent dynamics on detailed unstructured meshes, as featured in our WING datasets. This motivated the creation of these new datasets to provide a realistic and challenging application context.

All the simulations in our experiments are governed by the incompressible Navier-Stokes equations, which describe the motion of fluids and capture the balance of mass and momentum in incompressible flows. In their non-dimensional form, these equations are expressed as follows:

- Continuity equation (mass conservation):

$$\frac{\partial u}{\partial x} + \frac{\partial v}{\partial y} + \frac{\partial w}{\partial z} = 0$$

- Momentum conservation equations along the $x$, $y$, and $z$ directions, respectively:

$$\frac{\partial u}{\partial t} + u\frac{\partial u}{\partial x} + v\frac{\partial u}{\partial y} + w\frac{\partial u}{\partial z} = -\frac{\partial p}{\partial x} + \frac{1}{Re}\left(\frac{\partial^2 u}{\partial x^2} + \frac{\partial^2 u}{\partial y^2} + \frac{\partial^2 u}{\partial z^2}\right),$$

$$\frac{\partial v}{\partial t} + u\frac{\partial v}{\partial x} + v\frac{\partial v}{\partial y} + w\frac{\partial v}{\partial z} = -\frac{\partial p}{\partial y} + \frac{1}{Re}\left(\frac{\partial^2 v}{\partial x^2} + \frac{\partial^2 v}{\partial y^2} + \frac{\partial^2 v}{\partial z^2}\right),$$

$$\frac{\partial w}{\partial t} + u\frac{\partial w}{\partial x} + v\frac{\partial w}{\partial y} + w\frac{\partial w}{\partial z} = -\frac{\partial p}{\partial z} + \frac{1}{Re}\left(\frac{\partial^2 w}{\partial x^2} + \frac{\partial^2 w}{\partial y^2} + \frac{\partial^2 w}{\partial z^2}\right).$$

Here, $u(t, x, y, z)$, $v(t, x, y, z)$, and $w(t, x, y, z)$ represent the velocity components along the $x$, $y$, and $z$ directions, respectively; $p(t, x, y, z)$ denotes the pressure, and $Re$ is the Reynolds number, a dimensionless quantity measuring the ratio of inertial to viscous forces, which characterizes the flow regime (Pope, 2000).

Since these equations lack an analytical solution, our understanding of fluid behavior under different conditions relies heavily on experiments and numerical simulations. The flow regime can be classified based on the Reynolds number. For low Reynolds numbers, the flow is laminar, exhibiting smooth and predictable layers with minimal disruption. For high Reynolds numbers, the flow becomes turbulent, characterized by chaotic, irregular fluctuations. Although the Navier-Stokes

equations are deterministic, turbulent flows are chaotic due to the high nonlinearity of these equations, which renders the system sensitive to even minor perturbations. Despite this chaos, in fully developed turbulence, the energy introduced (e.g., from shear or pressure gradients) is balanced by energy dissipation through viscous forces, allowing the turbulence statistics to reach a steady state over time (Wilcox, 1998).

**ELLIPSEFLOW task** This task involves a canonical fluid dynamics problem, inferring the velocity $u$ and pressure $p$ fields in simulations of flow around an elliptical cylinder where the free-stream velocity is parallel to the ellipse's major axis (Figure 3b). The flow is laminar and unsteady, exhibiting periodic vortex shedding due to the formation of an unstable low-pressure wake behind the elliptical cylinder (Figure 3b). We compute probability density functions (PDFs) and statistical measures by sampling states within a periodic cycle.

In the training dataset (ELLIPSEFLOW-TRAIN), the Reynolds number ($Re$) ranges from 500 to 1000, and the ellipse's relative thickness ranges from 0.5 to 0.8. Test datasets evaluate model performance for in-distribution $Re$ and thickness (ELLIPSEFLOW-INDIST), as well as for lower and higher $Re$ (ELLIPSEFLOW-LOWRE and ELLIPSEFLOW-HIGHRE, respectively) and lower and higher thickness (ELLIPSEFLOW-THIN and ELLIPSEFLOW-THICK, respectively). These datasets are low-resolution versions of the datasets from Lino et al. (2022). Here, each simulation consists of 101 consecutive time-steps of fully developed 2D flow around an ellipse. The original simulations were generated using Nektar++ (Cantwell et al., 2015). We reduced the resolution by approximately a factor of two. The ellipse's major axis length is one (non-dimensional unit), and the domain height ranges from five to six. The training dataset contains 5000 simulations, while each test dataset comprises 50 simulations sampled from the original datasets in Lino et al. (2022). Additionally, the test dataset ELLIPSEFLOW-AOA contains 24 simulations for ellipses with an angle-of-attack (the angle between the ellipse's major axis and the free-stream velocity) of 10 degrees. Parameter values for each dataset are detailed in Table 2.

To condition probabilistic models, each node $i$ encodes the Reynolds number and a one-hot vector $\omega_i$ indicating its type (inlet, wall, or inner node), while each edge $(i, j)$ encodes the relative position between nodes, $e_{ij} := x_j - x_i$. The output is the velocity $u_i(t^*)$ and pressure $p_i(t^*)$ at each node $i$ and at an arbitrary time $t^*$.

Table 2: Values of the parameters for the ELLIPSE and ELLIPSEFLOW systems in the training and test datasets

| Dataset | Reynolds number $Re$ | Minor axis $b$ | Angle of attack (deg) | Average # nodes in ELLIPSE | Average # nodes in ELLIPSEFLOW | Purpose |
|---|---|---|---|---|---|---|
| ELLIPSE(FLOW)-TRAIN | [500, 1000] | [0.5 0.8] | 0 | 71 | 2340 | Training |
| ELLIPSE(FLOW)-INDIST | [500, 1000] | [0.5 0.8] | 0 | 70 | 2328 | Test |
| ELLIPSE(FLOW)-LOWRE | [400, 500] | [0.5 0.8] | 0 | 72 | 2376 | Test |
| ELLIPSE(FLOW)-HIGHRE | [1000, 1100] | [0.5 0.8] | 0 | 71 | 2312 | Test |
| ELLIPSE(FLOW)-THIN | [500, 1000] | [0.45 0.5] | 0 | 64 | 2286 | Test |
| ELLIPSE(FLOW)-THICK | [500, 1000] | [0.8 0.9] | 0 | 76 | 2282 | Test |
| ELLIPSE(FLOW)-AOA | [500, 1000] | [0.5 0.8] | 10 | 74 | 2514 | Test |

**ELLIPSE task** It involves inferring pressure $p$ on the wall of the ellipse from the previous task (Figure 3a). The pressure distribution on the ellipses in each ELLIPSEFLOW dataset was extracted to create these datasets. The Reynolds number is provided as an input condition for the probabilistic models at each node, and the relative position between adjacent nodes, along with the projection of the free-stream velocity along the edges are provided as input conditions for each edge. The models output the pressure $p_i(t^*)$ at each node $i$. The distribution of simulation parameters ($Re$ and relative thickness) in the ELLIPSE datasets mirrors that in the ELLIPSEFLOW datasets (Table 2).

In both ELLIPSE and ELLIPSEFLOW simulations, the temporal variance of the system's state increases primarily with the relative thickness of the ellipses due to the formation of larger vortices and, to a lesser extent, with $Re$. The effect of these parameters on temporal variance is shown in Figure 12a.

**WING task**   These experiments involve a wing in turbulent flow ($Re \sim 2 \times 10^6$), characterized by 3D vortices that spontaneously form and dissipate at various locations on the wing surface (Figure 3c). Each node $i$ encodes a unit outer normal vector $\hat{\boldsymbol{n}}_i$ and each edge $(i, j)$ encodes $\boldsymbol{e}_{ij} := [\boldsymbol{x}_j - \boldsymbol{x}_i, \boldsymbol{u}_\infty \cdot \hat{\boldsymbol{t}}_{ij}, \boldsymbol{u}_\infty \cdot \hat{\boldsymbol{n}}_i, \boldsymbol{u}_\infty \cdot \hat{\boldsymbol{b}}_{ij}]$, where $\boldsymbol{t}_{ij}$ is a unit vector parallel to the edge and $\hat{\boldsymbol{b}}_{ij} := \hat{\boldsymbol{t}}_{ij} \times \hat{\boldsymbol{n}}_i$. The output is $p_i(t^*)$. Unlike the previous systems, this flow is chaotic, and its time statistics converge over long time spans.

We generated the training and test datasets for the WING systems using the PISO solver of OpenFOAM with the Spalart-Allmaras Delayed Detached Eddy Simulation turbulence model (OpenFOAM Foundation, 2022). The simulation and dataset meshes were generated with snappyHexMesh. The free-stream velocity is 100 km/h, and the kinematic viscosity is $1.5 \times 10^{-5}$ m$^2$/s. A symmetry boundary condition was used on the vertical plane intersecting the wing root. The wing sections are 24XX NACA airfoils with constant relative thickness (indicated by XX), and the quarter-chord is horizontal (i.e., zero dihedral angle). The wing has a length of 1 m and a root chord length of 1 m, with a root-section angle-of-attack of 20 degrees. The chord length decreases linearly toward the tip, and the angle of attack also varies linearly toward the tip. The geometric parameters of the wing that vary across the training and test datasets include the relative thickness, the taper ratio (ratio between the tip and root chords), the sweep angle (angle between the quarter-chord line and a line perpendicular to the wing root), and the twist angle (angle of attack at the tip minus the angle of attack at the root). The values of these parameters for each dataset are listed in Table 3. Time-wise variance pri-

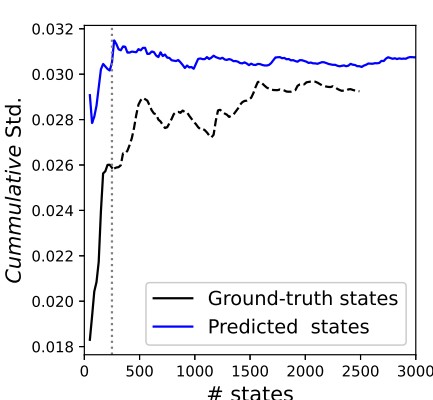

Figure 10: Node-wise mean of the time/sample-wise standard deviation computed over an increasing number of states. In the training trajectories, only the first 250 time-steps after the transient stage are used, and their standard deviation is not close to its ground-truth value, which is computed over 2,500 time-steps.

marily depends on the wing's relative thickness, with lower values resulting in higher variance due to more detached flow. Variance also increases with larger twist angles and smaller sweep angles.

The training dataset consists of 1,000 simulations, and the test dataset consists of 16 simulations. We simulated the first 4 seconds (transient regime) starting from a RANS solution, then recorded the pressure on the wing every 0.002 seconds for 0.5 seconds (250 time-steps) in the training dataset, and for 5 seconds (2,500 time-steps) in the test dataset. The first 250 time-steps are not sufficient to represent the variability of the states, as illustrated in Figure 10, but we aim to reduce the high cost of running these simulations by relying on the probabilistic models' ability to learn common patterns from an ensemble of simulations.

Table 3: Values of the parameters for the WING systems in the training and test datasets

| Dataset | Thickness | Taper ratio | Sweep (degrees) | Twist (degrees) | # time steps | Average # nodes | Purpose |
|---|---|---|---|---|---|---|---|
| WING-TRAIN | [10, 14] | [0.3, 0.7] | [20, 40] | [-5, 5] | 250 | 6852 | Training |
| WING-TEST | 11, 13 | 0.4, 0.6 | 25, 35 | -2.5, 2.5 | 2500 | 6828 | Testing |

The conditional features provided as input, as well as the predicted output for each system, are listed in Table 4.

# D   SUPPLEMENTARY RESULTS

Tables 5 and 6 collect our measures of sample accuracy, the coefficient of determination, on the test datasets; and Tables 7 to 9 collect our measures of distributional accuracy, the graph-level

Table 4: Conditional inputs and predicted outputs for each system type

| System | Node conditions $\boldsymbol{v}_i$ | Edge conditions $\boldsymbol{e}_{ij}$ | Node outputs $\boldsymbol{z}_i$ |
|---|---|---|---|
| ELLIPSE | $Re$ | $\boldsymbol{x}_j - \boldsymbol{x}_i, \boldsymbol{u}_\infty \cdot \hat{\boldsymbol{t}}_{ij}$ | $p_i$ |
| ELLIPSEFLOW | $Re, \boldsymbol{\omega}_i$ | $\boldsymbol{x}_j - \boldsymbol{x}_i$ | $u_i, v_i, p_i$ |
| WING | $\hat{\boldsymbol{n}}_i$ | $\boldsymbol{x}_j - \boldsymbol{x}_i, \boldsymbol{u}_\infty \cdot \hat{\boldsymbol{t}}_{ij}, \boldsymbol{u}_\infty \cdot \hat{\boldsymbol{n}}_{ij}, \boldsymbol{u}_\infty \cdot \hat{\boldsymbol{b}}_{ij}$ | $p_i$ |

Figure 11: For a system in dataset ELLIPSE-INDIST, probability density function from the DGN, LDGN, and baseline models trained on 100-time-step trajectories (full distribution). Ground truth in Figure 4b.

Wasserstein-2 distance. The correlation between the variance of the ground-truth trajectories and the accuracy of the LDGN's predictions is illustrated in Figure 12. Figures 13 and 14 showcase additional examples of results generated by DGN, LDGN, and baseline models for the ELLIPSEFLOW and WING tasks.

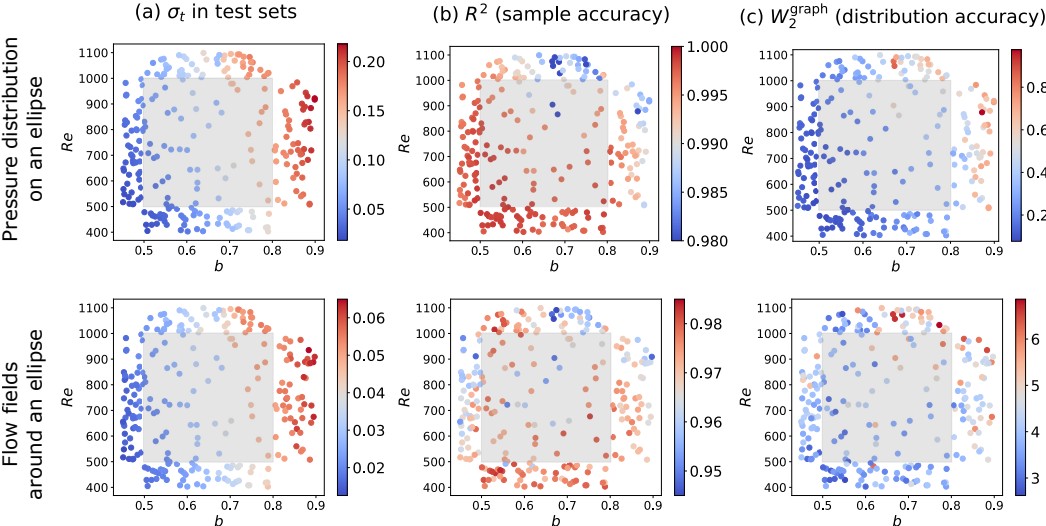

Figure 12: For each simulation in the ELLIPSE (top row) and ELLIPSEFLOW (bottom row) test datasets with a 0-degree angle-of-attack: (a) node-wise mean of the time-wise standard deviation of the ground truth fields, (b) average $R^2$ between the LDGN outputs and their corresponding ground truth, and (c) graph-wise $W_2$ distance between the learned distribution of the fields and their ground truth. Higher standard deviation results in more difficulty generating accurate fields and learning their probability distribution.

Table 5: Mean coefficient of determination ($R^2$) on the ELLIPSE datasets

| ELLIPSE / Model | -InDist | -LowRe | -HighRe | -Thin | -Thick | -AoA |
|---|---|---|---|---|---|---|
| Vanilla GNN | 0.984 ± 0.017 | 0.991 ± 0.007 | 0.974 ± 0.018 | 0.992 ± 0.003 | 0.964 ± 0.025 | 0.948 ± 0.029 |
| Bayesian GNN | 0.962 ± 0.025 | 0.967 ± 0.026 | 0.952 ± 0.021 | 0.952 ± 0.047 | 0.945 ± 0.020 | 0.931 ± 0.050 |
| GR-GNN | 0.944 ± 0.034 | 0.971 ± 0.018 | 0.902 ± 0.039 | 0.966 ± 0.017 | 0.900 ± 0.033 | 0.907 ± 0.031 |
| GM-GNN | 0.951 ± 0.030 | 0.973 ± 0.019 | 0.912 ± 0.033 | 0.970 ± 0.013 | 0.901 ± 0.032 | 0.922 ± 0.031 |
| VGAE | 0.981 ± 0.038 | 0.989 ± 0.026 | 0.971 ± 0.047 | 0.996 ± 0.006 | 0.947 ± 0.077 | 0.951 ± 0.049 |
| DGN | 0.994 ± 0.006 | 0.997 ± 0.001 | **0.988 ± 0.015** | 0.994 ± 0.002 | **0.992 ± 0.007** | **0.968 ± 0.026** |
| LDGN | **0.995 ± 0.007** | **0.998 ± 0.002** | 0.986 ± 0.019 | **0.997 ± 0.001** | 0.991 ± 0.009 | 0.966 ± 0.028 |

Table 6: Mean coefficient of determination ($R^2$) on the ELLIPSEFLOW datasets

| ELLIPSE FLOW / Model | -InDist | -LowRe | -HighRe | -Thin | -Thick | -AoA |
|---|---|---|---|---|---|---|
| Vanilla GNN | 0.979 ± 0.015 | 0.983 ± 0.011 | 0.972 ± 0.016 | 0.984 ± 0.007 | 0.962 ± 0.022 | 0.972 ± 0.019 |
| GM-GNN | 0.954 ± 0.022 | 0.964 ± 0.012 | 0.940 ± 0.021 | 0.967 ± 0.008 | 0.923 ± 0.032 | 0.947 ± 0.024 |
| VGAE | 0.981 ± 0.014 | 0.983 ± 0.012 | 0.974 ± 0.016 | 0.984 ± 0.008 | 0.966 ± 0.021 | 0.976 ± 0.014 |
| DGN | **0.990 ± 0.010** | **0.993 ± 0.007** | **0.982 ± 0.016** | **0.989 ± 0.009** | **0.991 ± 0.005** | **0.987 ± 0.014** |
| LDGN | 0.987 ± 0.013 | 0.992 ± 0.009 | 0.979 ± 0.017 | 0.986 ± 0.011 | 0.988 ± 0.007 | 0.981 ± 0.016 |

## D.1 PERFORMANCE

Providing a truly hardware-agnostic performance comparison is challenging. Most of the neural network components in our models are optimized for GPUs, while official OpenFOAM distributions lack GPU support. Additionally, performance results can vary depending on the target statistic (e.g., mean flow requires shorter simulations and less sampling compared to RMS calculations).

In Table 10, we present a breakdown of the runtime for the DGN and LDGN models on our WING experimental domain. The runtimes were measured on a CPU, limited to 8 threads, and on a single RTX 3080 GPU. The time for estimating the distribution accounts for the generation of 3,000 samples. This number of samples is sufficient for the RMS of the pressure to converge. When running on the GPU, we maximized concurrency to achieve 100% GPU utilization (50 to 60 concurrent samples). The ground-truth simulation was performed using OpenFOAM's PISO solver on the same workstation, limited to 8 CPU threads. Excluding pre-processing and post-processing steps, a nine-second simulation (required for the RMS to become statistically stationary) takes 50 hours.

Table 7: Mean $|\mathcal{V}|$-dimensional (i.e., graph-wise) Wasserstein-2 distance ($W_2^{\text{graph}}$) on the ELLIPSE datasets

| ELLIPSE / Model | -InDist | -LowRe | -HighRe | -Thin | -Thick | -AoA |
|---|---|---|---|---|---|---|
| Vanilla GNN | 0.96 ± 0.52 | 0.70 ± 0.44 | 1.32 ± 0.49 | 0.35 ± 0.10 | 2.15 ± 0.27 | 1.31 ± 0.35 |
| Bayesian GNN | 0.83 ± 0.35 | 0.66 ± 0.25 | 1.05 ± 0.34 | 0.49 ± 0.04 | 1.87 ± 0.22 | 1.28 ± 0.26 |
| GR-GNN | 1.09 ± 0.55 | 0.79 ± 0.42 | 1.49 ± 0.51 | 0.44 ± 0.12 | 2.32 ± 0.24 | 1.51 ± 0.53 |
| GM-GNN | 1.07 ± 0.52 | 0.77 ± 0.43 | 1.44 ± 0.51 | 0.43 ± 0.11 | 2.29 ± 0.24 | 1.38 ± 0.46 |
| VGAE | 0.44 ± 0.25 | 0.32 ± 0.21 | 0.59 ± 0.21 | 0.13 ± 0.03 | 1.13 ± 0.20 | 0.80 ± 0.16 |
| DGN | 0.29 ± 0.15 | 0.21 ± 0.09 | 0.42 ± 0.18 | 0.16 ± 0.02 | **0.56 ± 0.14** | **0.58 ± 0.1** |
| LDGN | **0.23 ± 0.12** | **0.17 ± 0.08** | **0.42 ± 0.18** | **0.10 ± 0.02** | 0.57 ± 0.15 | 0.59 ± 0.11 |

Table 8: Mean $|\mathcal{V}|$-dimensional (i.e., graph-wise) Wasserstein-2 distance ($W_2^{\text{graph}}$) on the ELLIPSE-FLOW datasets

| ELLIPSE FLOW / Model | -INDIST | -LOWRE | -HIGHRE | -THIN | -THICK | -AOA |
|---|---|---|---|---|---|---|
| Vanilla GNN | 6.23 ± 1.93 | 5.87 ± 1.61 | 6.88 ± 1.80 | 4.10 ± 0.68 | 9.73 ± 1.49 | 7.302 ± 1.700 |
| GM-GNN | 7.24 ± 2.12 | 6.76 ± 1.67 | 8.29 ± 2.04 | 5.05 ± 0.83 | 10.77 ± 1.69 | 8.393 ± 1.767 |
| VGAE | 6.10 ± 1.83 | 5.60 ± 1.54 | 6.70 ± 1.72 | 4.00 ± 0.68 | 9.36 ± 1.51 | 7.077 ± 1.607 |
| DGN | 4.72 ± 2.10 | 4.04 ± 1.74 | 5.48 ± 2.01 | 3.20 ± 0.81 | 7.76 ± 2.39 | 5.964 ± 2.125 |
| LDGN | **3.07 ± 0.93** | **2.53 ± 0.71** | **3.84 ± 1.24** | **2.81 ± 0.59** | **3.62 ± 0.91** | **3.709 ± 0.864** |

Table 9: Mean $|\mathcal{V}|$-dimensional (i.e., graph-wise) Wasserstein-2 distance ($W_2^{\text{graph}}$) on the WING-TEST dataset

| Metric | GM-GNN | DGN | LDGN |
|---|---|---|---|
| $W_2^{\text{graph}}$ | 4.32 ± 0.86 | 2.12 ± 0.90 | **1.95 ± 0.89** |

Compared to this baseline, we observe a one to two order-of-magnitude speedup on the CPU and a two to three order-of-magnitude speedup on the GPU. For single-sample generation, the speedup is indicated with respect to the time-stepping runtime of the numerical solver. This speedup would appear even more significant if we also accounted for the simulation of the transition regime.

The LDGN model is $8\times$ more efficient than the DGN due to performing the denoising process in a compressed latent space. However, this efficiency is only observed when the CPU/GPU cores are fully utilized.

## D.2 INFLUENCE OF THE NUMBER OF SCALES IN DGN MODELS

A key distinction of our DGN architecture, compared to previous GNN-based DDPM models, is the use of a U-Net-like architecture with graph pooling and unpooling layers specifically designed for mesh graphs (see Appendix B.3). Previous models were limited to small graphs, where applying message passing directly on the input graph was sufficient (Xu et al., 2022; Hoogeboom et al., 2022; Trippe et al., 2023; Vignac et al., 2023; Wu et al., 2024; Yi et al., 2024; Wen et al., 2023). However, DDPMs are known to require hierarchical and global denoising transitions, as these allow the model to capture and process information across multiple spatial resolutions (Dhariwal & Nichol, 2021; Si et al., 2023). Below, we compare our DGN with a single-scale model. For a fair comparison to heterogeneous previous work, we have considered a single-scale version of our DGN, with the

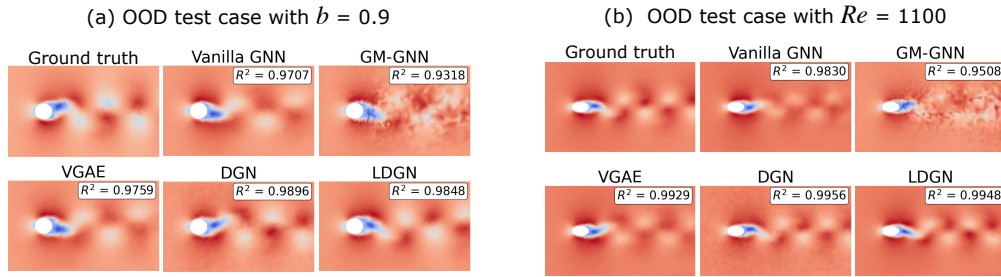

Figure 13: Samples from the DGN, LDGN, baseline models, and ground truth for (a) a simulation from dataset ELLIPSEFLOW-THICK, and (b) a simulation from dataset ELLIPSEFLOW-HIGHRE. The DGN and LDGN achieve the highest sample accuracy, with the DGN showing good accuracy but retaining some high-frequency noise.

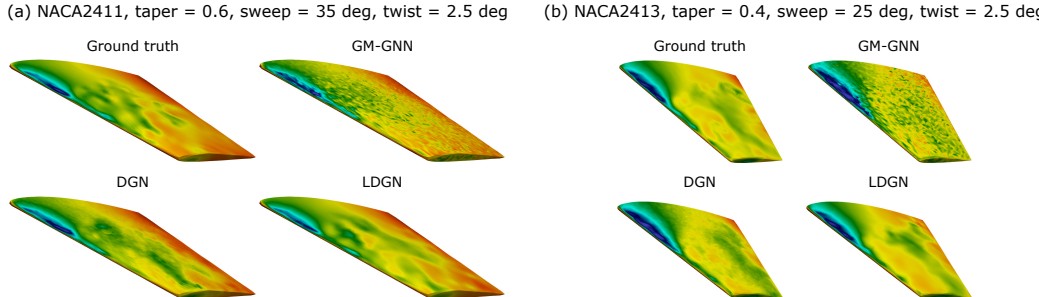

Figure 14: Samples from the DGN, LDGN, GM-GNN, and ground truth for two simulation from the WING-TEST dataset.

Table 10: Performance comparison for sample generation and distribution estimation on the WING task, along with the speedup achieved over the numerical solver

| Model | CPU s/sample | CPU min/distribution (speedup) | GPU s/sample | GPU min/distribution (speedup) |
|---|---|---|---|---|
| DGN | 6.81 | 340 (×8.8) | 0.59 | 19 (×152) |
| LDGN | 0.98 | 49 (×61) | 0.20 | 2.43 (×1235) |

same denoising parametrization, number and formulation of message-passing layers, diffusion-step conditioning, and training settings.

In the ELLIPSE task (with approximately 70 nodes per graph), a four-scale DGN significantly outperformed its single-scale counterpart (with the same total number of message-passing layers) in both sample and distributional accuracy, as shown in Table 11. In the ELLIPSEFLOW task (approximately 2.3k nodes per graph) and the WING task (approximately 6.8k nodes per graph), the single-scale DGN failed to produce feasible solutions, as illustrated in Figure 15. Instead, the model learned a transition noise close to zero, resulting in the amplification of the initial Gaussian noise during the denoising process, as described by equation (2). In the ELLIPSEFLOW task, performance improved as the number of scales increased (while maintaining the same total number of message-passing layers), reaching optimal results with the five-scale model, as also shown in Table 11. This five-scale DGN was also the most computationally efficient: it was 25% faster than the two-scale model, 12% faster than the three-scale model, and 7% faster than the four-scale model.

Table 11: Sample accuracy ($R^2$), distributional accuracy ($W_2^{\text{graph}}$) and inference time for GNN models with varying numbers of scales

| Dataset | # scales in model | $R^2$ | $W_2^{\text{graph}}$ | Inference time ms /sample |
|---|---|---|---|---|
| ELLIPSE-INDIST | 1 | 0.964 ± 0.043 | 0.60 ± 0.28 | 204 |
| | 4 | **0.995 ± 0.006** | **0.29 ± 0.15** | **160** |
| ELLIPSEFLOW-INDIST | 2 | 0.957 ± 0.026 | 5.37 ± 1.34 | 436 |
| | 3 | 0.966 ± 0.034 | 5.48 ± 2.04 | 376 |
| | 4 | **0.990 ± 0.010** | 5.01 ± 2.10 | 353 |
| | 5 | **0.990 ± 0.010** | **4.72 ± 2.11** | **328** |

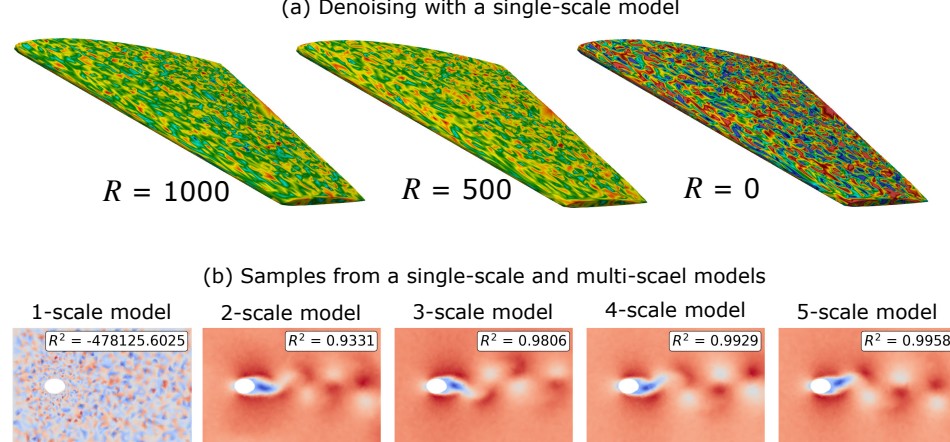

Figure 15: Single-scale models failed to learn in the ELLIPSEFLOW and WING tasks due to the limited range of their receptive fields. (a) Instead of removing noise between transitions, a single-scale model amplified the initial Gaussian noise. (b) Samples from DGN models with one-, two-, three-, four-, and five-scale architectures for a simulation from the ELLIPSEFLOW-INDIST dataset.

### D.3 ALTERNATIVE VGAE ARCHITECTURE

Our VGAE incorporates a condition encoder block (Figure 9a) that generates an encoding of $V_c$ and $E_c$ on the latent graph $\mathcal{G}^L$, which is then passed as a conditioning input to the LDGN model. These encodings cannot be directly generated by the VGAE's node encoder (Figure 9b), as the node encoder also requires $Z(t)$ as input, which is unavailable during inference since we directly sample Gaussian noise in the latent space.

An alternative approach to ours could be to define the LDGN conditions directly within the coarse representation of the system provided by $\mathcal{G}^L$. To test this, we compared our LDGN with another version that lacks the condition encoder in the VGAE (but has similar number of weights), where node and edge conditional features are defined directly on $\mathcal{G}^L$. We evaluated this in the ELLIPSE task and found that our proposed LDGN architecture provides better sample and distributional accuracy, as shown in Tables 12 and 13. This improvement is likely because our conditional encodings, although compressed, still retain information about relevant high-frequency features.

Table 12: Comparison of the coefficient of determination ($R^2$) for an LDGN using our VGAE and one using the alternative VGAE

| ELLIPSE
Model | -INDIST | -LOWRE | -HIGHRE | -THIN | -THICK |
|---|---|---|---|---|---|
| Alternative | 0.993 ± 0.009 | 0.996 ± 0.003 | 0.985 ± 0.017 | 0.994 ± 0.004 | 0.990 ± 0.010 |
| Ours | **0.996 ± 0.007** | **0.998 ± 0.002** | **0.988 ± 0.018** | **0.998 ± 0.001** | **0.993 ± 0.008** |

Table 13: Comparison of the graph-wise Wasserstein-2 distance ($W_2^{\text{graph}}$) for an LDGN using our VGAE and one using the alternative VGAE

| ELLIPSE
Model | -INDIST | -LOWRE | -HIGHRE | -THIN | -THICK |
|---|---|---|---|---|---|
| Alternative | 0.22 ± 0.08 | 0.44 ± 0.17 | 0.16 ± 0.03 | 0.60 ± 0.16 | 0.62 ± 0.09 |
| Ours | **0.17 ± 0.07** | **0.41 ± 0.17** | **0.10 ± 0.02** | **0.54 ± 0.16** | **0.57 ± 0.10** |

## D.4 RESULTS FOR HIGH-FREQUENCY ERROR ANALYSIS

To quantify the high-frequency noise present in the DGN and LDGN samples, we computed the graph Fourier transform of the predicted fields and assessed the energy associated with the 10 highest graph eigenvalues for all systems in the ELLIPSEFLOW-INDIST and WING-TEST datasets. This serves as a measure of the high-frequency noise present in the predicted fields. The error of this high-frequency energy relative to its ground truth, in percent, is collected in Table 14.

We note that the high-frequency noise in the samples generated by DGN models is consistently higher than in the ground-truth fields. In contrast, the noise in LDGN samples differs only slightly.

Table 14: Relative error (%) between the high-frequency energy in the predicted samples and the ground truth

| Dataset & Magnitude | | DGN | LDGN |
|---|---|---|---|
| | $u$ | 3.38 | 1.87 |
| ELLIPSEFLOW-INDIST | $v$ | 8.97 | 3.49 |
| | $p$ | 11.09 | 3.11 |
| WING-TEST | $p$ | 2.84 | 0.97 |

## D.5 EFFICIENCY IN LEVERAGING TRAINING TRAJECTORIES

While the results in Section 5 demonstrate the efficacy of our approach, there is no theoretical guarantee that a model can learn *full* distributions from a collection of short trajectories. This ability relies on the model's capacity to recognize common physical patterns across different trajectories and leverage this knowledge to enhance its understanding of each individual trajectory. Such interpolative capacity depends on two key factors: (i) having a sufficient number of training trajectories to enable smooth interpolation across the parameter space, and (ii) the model's ability to effectively capture and generalize these dependencies.

To disentangle the effects of the number of training trajectories and the model's design, we conducted additional experiments on the ELLIPSE task. Specifically, we trained DGNs, LDGNs, and baseline models on datasets containing varying numbers of short trajectories, ranging from 500 to 5,000 samples. As shown in Figure 16, the distributional accuracy of the DGN and LDGN models improves with an increasing number of training trajectories, demonstrating their ability to interpolate effectively across trajectories associated with different input conditions and accurately reproduce the underlying distribution. In contrast, the best-performing baseline model (VGAE) plateaued in performance beyond 2,000 trajectories, highlighting its limited interpolation capacity despite having the same number of learnable parameters.

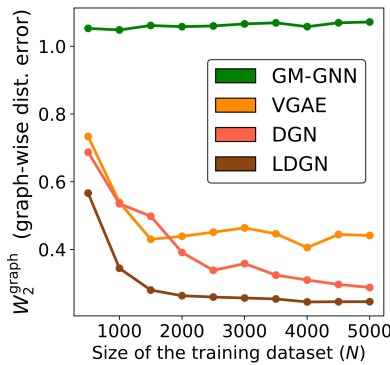

Figure 16: The DGN, and especially the LDGN, models are more efficient than the VGAE and GM-GNN at learning common features from a collection of short trajectories, resulting in significantly lower distributional error in large datasets.

This advantage likely stems from the diffusion-based approach of DGNs and LDGNs, which, unlike models that rely on sampling from a compressed latent space (e.g., VGAE) or impose assumptions about the underlying probability distributions (e.g., GM-GNN), minimizes overfitting by gradually denoising. This approach enables them to infer missing dynamics by leveraging similar states encountered during training. Furthermore, the LDGN variant outperforms the DGN by learning distributions in a compressed latent space that is less sensitive to small-scale fluctuations, resulting in superior interpolation performance.

Regarding sampling accuracy, we verified that it remains unaffected by the number of training trajectories. This behavior aligns with our expectations, as accurate samples can still be generated even without the ability to complete the system's PDF by feature interpolation.

### D.6 FLOW STATISTICS DERIVED FROM LEARNED DISTRIBUTIONS

With our DGN and LDGN models, we aim to learn the probability density function (PDF) of fully-developed unsteady flows. In certain cases, however, learning the mean flow alone using a deterministic GNN trained directly on mean-flow data might suffice. Nevertheless, such deterministic models may encounter challenges when the available training trajectories are too short to properly capture the mean flow. Furthermore, the distributions generated by the DGN and LDGN models can be used to derive other relevant statistics, such as the turbulent kinetic energy (TKE) or the Reynolds shear stress.

To evaluate the advantage provided by the DGN and LDGN models over deterministic baselines in the mean-flow prediction task, we trained a deterministic model, referred to as the *Mean-Flow GNN*, on 10-time-step-averaged flow fields from the ELLIPSEFLOW-TRAIN dataset. We compared its performance to that of the models discussed in Section 5: the Vanilla GNN (another deterministic model trained to maximize the likelihood; see Section B.5), the DGN, the LDGN, and other probabilistic baselines. The coefficient of determination for mean-flow predictions, as evaluated on the ELLIPSEFLOW test datasets, is summarized in Table 15. Notably, the LDGN significantly outperforms both the deterministic and probabilistic baselines in predicting the mean flow, achieving coefficient of determination values close to one – despite being trained on only 10 consecutive states per trajectory.

Figure 17 further illustrates these findings. Predictions from the two deterministic baselines (Mean-Flow GNN and Vanilla GNN) exhibit diffused vertex distributions corresponding to the 10-time-step (training simulation length) average fields. This diffusion indicates that during training, these models are overly biased toward the conditions of individual samples and fail to generalize the true average flow across training samples. The DGN slightly mitigates this problem through its gradual denoising approach, which minimizes overfitting to individual training samples. The improvement is more evident in the LDGN, which, thanks to its more meaningful latent space and latent input conditions, better leverages the information present across systems that are close in parameter space.

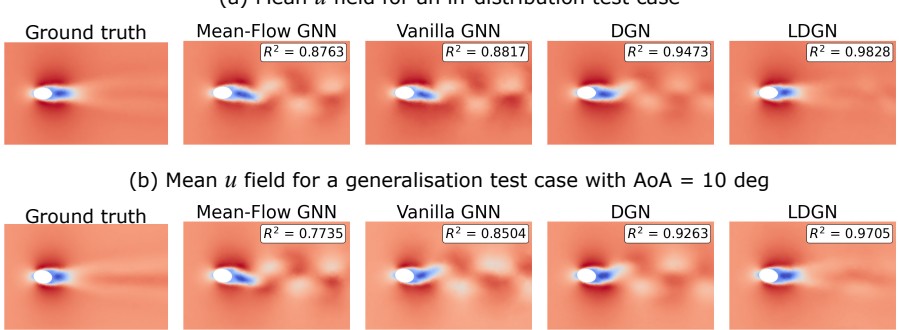

Figure 17: Mean $u$ field predicted by the Mean-Flow GNN and the Vanilla GNN (deterministic GNNs), as well as obtained from the distributions of the DGN and LDGN, for a test case from (a) dataset ELLIPSEFLOW-INDIST and (b) dataset ELLIPSEFLOW-AOA.

To further demonstrate the effectiveness of the proposed diffusion models, we computed the turbulent kinetic energy (TKE). This quantity is defined as:

$$\text{TKE} = \frac{1}{2} \left( \langle u'u' \rangle + \langle v'v' \rangle \right)$$

where $\langle \cdot \rangle$ denotes ensemble averaging, and $u' := u - \langle u \rangle$ and $v' := v - \langle v \rangle$ are velocity fluctuations. The coefficients of determination for the TKE are presented in Table 16. Reflecting the superior distributional accuracy of the LDGN, this model significantly outperformed the baseline models, with the DGN emerging as the second-best performer. The LDGN's advantage in these predictions

Table 15: Mean coefficient of determination ($R^2$) of the mean-flow on the ELLIPSEFLOW datasets

| ELLIPSE FLOW / Model | -INDIST | -LOWRE | -HIGHRE | -THIN | -THICK | -AOA |
|---|---|---|---|---|---|---|
| Mean-Flow GNN | 0.868 ± 0.014 | 0.872 ± 0.012 | 0.865 ± 0.015 | 0.883 ± 0.002 | 0.843 ± 0.008 | 0.863 ± 0.013 |
| Vanilla GNN | 0.869 ± 0.013 | 0.871 ± 0.012 | 0.864 ± 0.014 | 0.883 ± 0.003 | 0.845 ± 0.009 | 0.864 ± 0.013 |
| GM-GNN | 0.867 ± 0.015 | 0.870 ± 0.012 | 0.857 ± 0.016 | 0.883 ± 0.003 | 0.841 ± 0.009 | 0.860 ± 0.014 |
| VGAE | 0.926 ± 0.014 | 0.931 ± 0.012 | 0.921 ± 0.015 | 0.942 ± 0.003 | 0.902 ± 0.011 | 0.921 ± 0.015 |
| DGN | 0.941 ± 0.020 | 0.946 ± 0.015 | 0.936 ± 0.021 | 0.954 ± 0.006 | 0.912 ± 0.028 | 0.930 ± 0.025 |
| LDGN | **0.976 ± 0.004** | **0.978 ± 0.002** | **0.975 ± 0.005** | **0.978 ± 0.001** | **0.974 ± 0.003** | **0.975 ± 0.003** |

is further illustrated in Figure 18. It is important to highlight that all these models were trained on only 10 consecutive time steps from each training simulation (see Section 5), making the results even more remarkable.

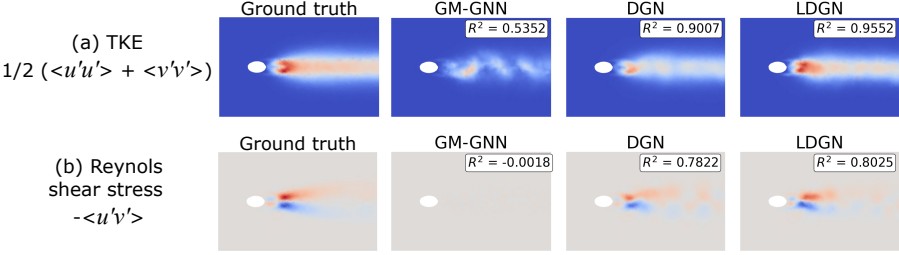

Figure 18: (a) Turbulent kinetic energy and (b) Reynolds shear stress obtained from the distributions predicted by the GM-GNN, DGN, and LDGN for a test case from the ELLIPSEFLOW-INDIST dataset.

Table 16: Mean coefficient of determination ($R^2$) of the TKE on the ELLIPSEFLOW datasets

| ELLIPSE FLOW / Model | -INDIST | -LOWRE | -HIGHRE | -THIN | -THICK | -AOA |
|---|---|---|---|---|---|---|
| GM-GNN | 0.669 ± 0.141 | 0.675 ± 0.155 | 0.646 ± 0.120 | 0.881 ± 0.043 | 0.393 ± 0.091 | 0.564 ± 0.132 |
| VGAE | -0.201 ± 0.040 | -0.182 ± 0.040 | -0.218 ± 0.050 | -0.145 ± 0.011 | -0.253 ± 0.038 | -0.197 ± 0.043 |
| DGN | 0.783 ± 0.215 | 0.874 ± 0.141 | 0.755 ± 0.241 | 0.863 ± 0.150 | 0.485 ± 0.334 | 0.654 ± 0.279 |
| LDGN | **0.911 ± 0.226** | **0.947 ± 0.064** | **0.876 ± 0.234** | **0.895 ± 0.071** | **0.901 ± 0.046** | **0.895 ± 0.156** |

## D.7 FAST SAMPLING WITH FLOW MATCHING

Flow matching (Lipman et al., 2023) has recently emerged as a promising generative model framework, offering significant efficiency gains over diffusion models by defining a direct linear mapping between samples of a Gaussian distribution (or any other distribution, in theory) and the target distribution. This enables sampling to be performed with fewer steps.

Our DGN and LDGN architectures can be seamlessly adapted to the flow-matching training framework, benefiting from faster sampling. We observed that flow-matching GNNs (FM-GNNs) and latent-flow-matching GNNs (LFM-GNNs) outperform their diffusion-based counterparts when the number of denoising steps is limited to 10 or fewer. However, for ∼20 or more denoising steps, diffusion models demonstrate superior performance.

This behavior is illustrated in Figure 19a, which compares the standard deviation of the pressure predicted by the DGN and the FM-GNN with 10 denoising steps for a sample from the WING-TEST dataset (ground truth shown in Figure 6). The shift in performance for 20 or more denoising steps is

shown in Figure 19b, which presents the sample accuracy and distributional error as a function of the number of denoising steps for the DGN, FM-GNN, and their latent variants on the ELLIPSE-INDIST dataset.

From these results, we observe that the LFM-GNN achieves a compelling balance between accuracy and sampling speed. However, further experimentation is necessary to fully explore its potential, which we leave for future work.

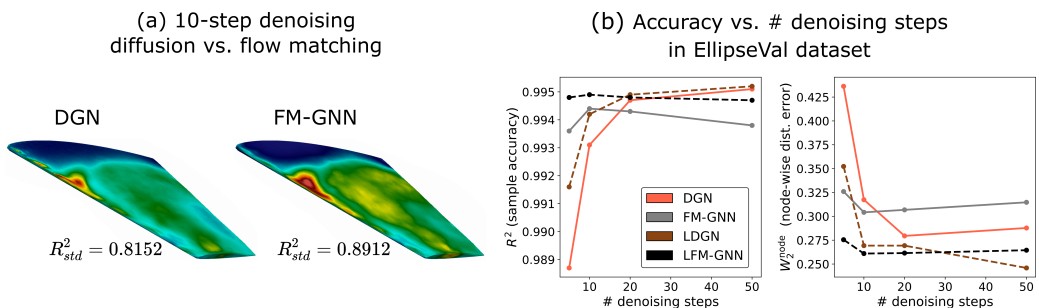

Figure 19: Flow-matching-based models outperform diffusion-based models when only a small number of denoising steps are employed for sampling. This was tested on (a) the WING-TEST dataset and (b) the ELLIPSE-INDIST dataset.

