# OpenReview forum: "Learning Distributions of Complex Fluid Simulations with Diffusion Graph Networks"
_ICLR.cc/2025/Conference — ICLR 2025 Oral_

### Official Review · Reviewer_uXMJ · 2024-10-29

**Soundness:** 4
**Presentation:** 4
**Contribution:** 3
**Rating:** 8
**Confidence:** 2

**Summary:**

This paper proposes using graph-based diffusion methods to sample steady states in fluid dynamics problems.  Impressively, the training does not need simulations run all the way to convergence.  Behind the scenes, the proposed model uses a multiscale graph network, downsampled using strategies that are typical in numerical PDE.

**Strengths:**

The results in this paper are impressive, and the algorithm makes sense.  I am particularly glad to see relatively realistic test cases as well as reasonable engineering justification for the design behind the proposed architecture.

Also, thank you for preparing a polished, easy-to-read paper submission.  Relative to the other ICLR papers I had to read, this one had a thoughtful and clear discussion.

**Weaknesses:**

From an algorithmic perspective, the building blocks of the proposed method are unsurprising and reflective of larger trends in the fields (graph networks with a multiscale aspect, diffusion sampling, and so on).  So, I would not consider this work to be too exciting from a methodological perspective, although perhaps a reader with more experience specifically in numerical fluid dynamics would better be able to evaluate this aspect.  I am willing to overlook this, however, as the method seems solidly engineered (why invent a new approach to ML just to claim novelty?) and the results look great.

**Questions:**

Line 107, “defined their discretization mesh” (typo?)

Can the authors provide intuition on why the problem described on line 142 is well posed?  In particular, if the learning method exclusively sees short trajectories that end before reaching equilibrium, why should this information be sufficient to train a method that directly predicts converged states it hasn’t seen in the training data?  The results seem to indicate this is possible, but it seems counterintuitive to me that this task is reasonable.

Line 169, “interpolates between the bounds of the process’ entropy” --- I wasn’t sure what this means.  Also, (Ho et al., 2020) isn’t written for graph-based models --- does this make a difference?

The receptive field of a “vanilla” DGN (line 180) is relatively small --- typically a vertex only “talks to” its k-ring, where k is the number of layers.  Does this affect performance of the proposed model, even after incorporating the latent diffusion method?

Line 207:  What does “perceptually equivariant” mean?  How do you verify?

Line 249:  no need for a hyphen in “bottleneck”

Line 384:  Do statistical limitations get in the way of estimating W_2^graph accurately?  It seems like this would be a high-dimensional optimal transport problem, which itself can be difficult to estimate reliably from samples.  How many samples are used here?

Line 451:  If I understand properly, the “3x faster” claim here would be substantially dampened if you had to include training time of the diffusion model.  Is this right?

---

> ### Author Response · Authors · 2024-11-19
>
> We appreciate the reviewer’s understanding of our approach and their positive assessment of our results. Our aim was indeed to focus on performance and applicability in this domain.
>
> **Q**: Can the authors provide intuition on why the problem described in line 142 is well-posed?
>
> **A**: The reviewer is correct that the problem, as presented, is not necessarily well-posed; there is no guarantee that a model can fully “learn” the probability distribution function (PDF) from only short trajectories. A more accurate phrasing would be that we aim to “approximate” the distribution by “interpolation” between short trajectories. Achieving this requires two key conditions: (1) sufficient training trajectories to enable smooth interpolation in the parameters space, and (2) a model capable of capturing these dependencies effectively.
>
> To investigate separately the role of the number of training trajectories and the role played by the model, we conducted additional experiments for the Ellipse task, training DGNs, LDGNs, and baseline models with different numbers of short trajectories. We observed that as the number of training trajectories is increased the distributional accuracy of the DGN and LDGN models improves, indicating that they can effectively interpolate across parameter space. By contrast, the best baseline model (VGAE) plateaued in performance for more than 2000 trajectories, indicating a limited capacity for interpolation. We also observed that DGN and LDGN still outperform the baselines for any given dataset length.
>
> In our original results section, we already suggested that the diffusion-based models’ success in learning the full distribution from short trajectories likely arises from their capacity to recognize shared physical patterns across systems. We will include these new results in the revised appendix, clarify this in the Results section, and update the method section to explicitly state our assumptions.
>
> **Q**: Line 169, “interpolates between the bounds of the process’ entropy” --- I wasn’t sure what this means. Also, (Ho et al., 2020) isn’t written for graph-based models --- does this make a difference?
>
> **A**: Here, $\beta_r$ and $\tilde{\beta}_r$ denote the bounds of the process’ entropy. This citation was referring to the mathematical definition for the bounds of the process’ entropy given in that paper. We have modified this for more clarity.
>
> **Q**: The receptive field of a “vanilla” DGN (line 180) is relatively small --- typically a vertex only “talks to” its k-ring, where k is the number of layers. Does this affect performance of the proposed model, even after incorporating the latent diffusion method?
>
> **A**: To clarify, in our paper, “vanilla” GNN refers to a deterministic GNN, while the reviewer likely refers to the “flat” or single-scale DGN. We have not tested an LDGN with a single-scale GNN operating in the latent space. However, even such a configuration could arguably not be considered truly single-scale, as the autoencoder still employs a multi-scale GNN for latent compression.
> Testing a single-scale GNN in the latent space could be an interesting avenue for exploration. We hypothesize that such a model would struggle with large systems (e.g., EllipseFlow and Wing tasks) unless the latent space is reduced to a very small number of nodes. In that scenario, the autoencoder would have to take on much of the generative burden, likely leading to degraded performance closer to that of the baseline VGAE. Other architectures with a global receptive field, such as transformers, could be interesting alternatives for the latent space processing. We will discuss this as a direction for future work in the revised manuscript.

---

> ### Author Response · Authors · 2024-11-19
>
> **Q**: What does “perceptually equivariant” mean? How do you verify?
> A: Thank you for pointing this out. This is indeed a typo; it should read “perceptually equivalent” as discussed in [Rombach et al., 2022]. Perceptual equivalence means that the latent representation preserves nearly all the information necessary to reconstruct the original’s perceptual qualities. This is typically achieved by training the latent space using an autoencoder. For more details on the distinction between perceptual and semantic compression, we refer the reviewer to [Rombach et al., 2022].
>
> **Q**: Do statistical limitations get in the way of estimating $W_2^graph$ accurately? It seems like this would be a high-dimensional optimal transport problem, which itself can be difficult to estimate reliably from samples. How many samples are used here?
>
> **A**: ​​You are correct that estimating $W_2^{\text{graph}}$ accurately can become challenging for high-dimensional problems, as it depends on the dimensionality and complexity of the data. In our experiments, we use different numbers of samples depending on the task:
>
> - For the Ellipse and EllipseFlow tasks, the ground-truth distributions are represented by 100 evenly distributed samples, while the predicted distributions are represented by 200 samples.
>
> - For the Wing task, we use 2500 samples for the ground-truth distributions and 3000 samples for the predicted distributions.
> While the computational cost for evaluating the $W_2^{\text{graph}}$ distance increases with the number of samples and dimensions, this level of sampling is necessary for a reliable comparisons. We will clarify the number of samples used for each task in the revised manuscript.
>
> **Q**: If I understand properly, the “3x faster” claim here would be substantially dampened if you had to include the training time of the diffusion model. Is this right?
>
> **A**: The LDGN is approximately 3x faster than the DGN during sample inference, but its training time is 53% longer because two components must be trained: the autoencoder (which accounts for 60% of the total training time) and the latent diffusion model (40%). However, we consider fast inference to be more critical for models intended for design tasks, where they are evaluated repeatedly. In the revised manuscript, we will explicitly clarify that the accuracy and efficiency benefits of the LDGN come at the cost of an increased training time.

---

> > ### Comment · Reviewer_uXMJ · 2024-11-23
> >
> > Thanks for the clarifications!  Already my score is quite high.
> >
> > I would encourage the authors to include these clarifications in the text of the paper instead of just the rebuttal.

---

### Official Review · Reviewer_84S9 · 2024-10-30

**Soundness:** 3
**Presentation:** 2
**Contribution:** 2
**Rating:** 6
**Confidence:** 2

**Summary:**

The paper aims to predict possible steady states of complex unsteady dynamical systems given their mesh discretization and physical parameters. With a novel graph-based (latent) diffusion model to predict equilibrium states of dynamical systems, the proposed model can handle unstructured mesh.

**Strengths:**

- Integrating GNN with a diffusion model is an interesting idea, as it allows for better generality and great flexibility in unstructured mesh.
- The paper claims to predict a full distribution of possible equilibrium states, although trained on states of a short trajectory.

**Weaknesses:**

Some critical information is missing.
- The underlying physics problem needs to be explained more. I would expect some rigorous mathematical definitions such as PDEs to formulate the problem.
- The related work should cover more classical approaches to the problem of predicting equilibrium states of dynamical systems.
- The learning problem should also be better formulated.

**Questions:**

- What is a statistical equilibrium?
- What is the generality of the model? Does it generalize to different trajectories, different physical parameters, or even different domain shapes or sizes?
- Generally, how complex a dynamical system is the model able to handle? Some dynamical systems such as the Lorenz system are chaotic, so are they still predictable with the proposed model?

---
**Update**

Thank you for your detailed response. I think I have a better understanding of the concepts, and I believe it is meaningful work. Thus, I decided to raise my score.

---

> ### Author Response · Authors · 2024-11-19
>
> We appreciate the reviewer’s feedback, which is being considered and incorporated to improve the clarity and quality of our paper. Below, we provide our detailed responses to each of the points raised.
>
> **Q**: The underlying physics problem needs to be explained more. I would expect some rigorous mathematical definitions such as PDEs to formulate the problem.
>
> **A**: We thank the reviewer for raising this, since researchers from outside the fluid-dynamics community may not be familiar with the Navier-Stokes equations and the challenges they pose.
>
> The simulations in the Ellipse, EllipseFlow, and Wing tasks are governed by the incompressible Navier-Stokes equations, which describe the motion of fluids and capture the balance of mass and momentum in incompressible flows. In their non-dimensional form, these equations are expressed as follows:
> Continuity equation (mass conservation):
> $$\frac{\partial u}{\partial x} + \frac{\partial v}{\partial y} + \frac{\partial w}{\partial z} = 0$$
> Momentum conservation equations along the $x$, $y$, and $z$ directions, respectively:
>
> $$\frac{\partial u}{\partial t} + u \frac{\partial u}{\partial x} + v \frac{\partial u}{\partial y} + w \frac{\partial u}{\partial z} = -\frac{\partial p}{\partial x} + \frac{1}{Re} \left( \frac{\partial^2 u}{\partial x^2} + \frac{\partial^2 u}{\partial y^2} + \frac{\partial^2 u}{\partial z^2} \right),$$
>
> $$\frac{\partial v}{\partial t} + u \frac{\partial v}{\partial x} + v \frac{\partial v}{\partial y} + w \frac{\partial v}{\partial z} = -\frac{\partial p}{\partial y} + \frac{1}{Re} \left( \frac{\partial^2 v}{\partial x^2} + \frac{\partial^2 v}{\partial y^2} + \frac{\partial^2 v}{\partial z^2} \right),$$
>
> $$\frac{\partial w}{\partial t} + u \frac{\partial w}{\partial x} + v \frac{\partial w}{\partial y} + w \frac{\partial w}{\partial z} = -\frac{\partial p}{\partial z} + \frac{1}{Re} \left( \frac{\partial^2 w}{\partial x^2} + \frac{\partial^2 w}{\partial y^2} + \frac{\partial^2 w}{\partial z^2} \right).$$
>
> Here, $u(t,x,y,z)$, $v(t,x,y,z)$, and $w(t,x,y,z)$ represent the velocity components along the $x$, $y$, and $z$ directions, respectively; $p(t,x,y,z)$ denotes the pressure, and $Re$ is the Reynolds number, a dimensionless quantity measuring the ratio of inertial to viscous forces, which characterizes the flow regime [Pope 2000].
>
> Since these equations lack an analytical solution, our understanding of fluid behavior under different conditions relies heavily on experiments and numerical simulations. The flow regime can be classified based on the Reynolds number:
> For low Reynolds numbers, the flow is laminar, exhibiting smooth and predictable layers with minimal disruption.
> For high Reynolds numbers, the flow becomes turbulent, characterized by chaotic, irregular fluctuations. Although the Navier-Stokes equations are deterministic, turbulent flows are chaotic due to the high nonlinearity of these equations, which renders the system sensitive to even minor perturbations. Despite this chaos, in fully developed turbulence, the energy introduced (e.g., from shear or pressure gradients) is balanced by energy dissipation through viscous forces, allowing the turbulence statistics to reach a steady state over time [Wilcox 1998, Moon].
> In our specific tasks:
>
> - Ellipse and EllipseFlow tasks ($400 \leq Re \leq 1100$): The flow is laminar, making it deterministic. Here, the fully-developed flow is unsteady and exhibits periodic vortex shedding due to the formation of an unstable low-pressure wake behind the elliptical cylinder (Figure 3b). We compute probability density functions (PDFs) and statistical measures by sampling states within a periodic cycle.
>
> - Wing task ($Re \sim 2 \times 10^6$): The flow is turbulent, resulting in chaotic behavior. In this task, we consider fully developed turbulence and calculate ground-truth statistics by integrating over long time intervals to achieve statistically converged results.
>
> We will include this explanation in Appendix C and point out the mentioned challenges in Section 4 so that readers from outside the fluid-dynamics community better understand the underlying problem and its relevance.

---

> ### Author Response · Authors · 2024-11-19
>
> **Q**: The related work should cover more classical approaches to the problem of predicting equilibrium states of dynamical systems.
>
> **A**: We appreciate the reviewer’s suggestion. It is important to clarify that we are not focusing on “strict” equilibrium states, where $\partial u_i / \partial t = 0$, but rather on statistical equilibrium states. In this context, while the magnitudes of the system remain time-dependent, their statistical measures (such as mean, standard deviation, and higher-order moments) are time-invariant. Predicting these statistical equilibrium states is a complex challenge due to the vast number of possible solutions and the inherent randomness of the system. Therefore, our work focuses on modeling their probability distribution. In many applications, the primary interest lies not in the mean flow but in the distributions of flow fields at equilibrium. For instance, RMS fluctuations are critical in the design of airfoils and turbulence research fundamentally revolves around flow field statistics (see Introduction section for References).
>
> While classical approaches have been developed to address aspects of this problem, most are domain-specific and tend to focus on specific statistical measures. For instance, as mentioned briefly in the introduction, Reynolds-Averaged Navier-Stokes (RANS) models can predict averaged flow solutions [Pope2000]. Another classical approach is the sum-of-squares-of-polynomials technique, which can derive upper and lower bounds for certain fluid quantities [1]. However, these methods are generally limited in their ability to capture the full range of statistical behavior in a system.
> For a more comprehensive representation of the probability distribution functions (PDFs) of the flow variables, we typically resort to Direct Numerical Simulations (DNS) or Large Eddy Simulations (LES), which are computationally expensive and require long time integration to obtain accurate statistical estimates.
>
> We will emphasize this in the Related Work section to highlight the relevance of our work and the potential of deep-learning-based probabilistic models.
>
> [1] Chernyshenko, S.I., Goulart, P., Huang, D. and Papachristodoulou, A., 2014. Polynomial sum of squares in fluid dynamics: a review with a look ahead. Philosophical Transactions of the Royal Society A: Mathematical, Physical and Engineering Sciences.
>
> **Q**: The learning problem should also be better formulated.
>
> **A**: In the revised version, in Section 3, we will clarify what is meant by statistical equilibrium and the learning objective.
>
> **Q**: What is a statistical equilibrium?
>
> **A**: As mentioned in the paper, “We assume the system reaches a converged statistical equilibrium after a transient phase, allowing stationary statistics to be computed over sufficiently long intervals.” In practical terms, statistical equilibrium means that the statistical measures of the system (such as the mean, standard deviation, or higher-order moments) remain constant over time. This is essential to ensure that the results are meaningful and reproducible.
>
> More specifically, statistical equilibrium refers to a state in which the statistical properties of a dynamic system are stable over time. This does not imply that the system is steady; rather, it means that its statistical measures, computed over sufficiently long time intervals, have reached a time-invariant state  [Wilcox1998]. Even if the system’s dynamics exhibit oscillatory behavior (as seen in the Ellipse and EllipseFlow simulations) or chaotic behavior (as seen in the Wing simulations), the statistical properties stabilize.
>
> We will make this clearer in Section 3, where we introduce our assumption of statistical equilibrium.

---

> ### Author Response · Authors · 2024-11-19
>
> **Q**: What is the generality of the model? Does it generalize to different trajectories, different physical parameters, or even different domain shapes or sizes?
>
> **A**: Different trajectories can result from varying domain geometries, boundary conditions, or physical parameters. We assume the reviewer is specifically asking about out-of-distribution (OOD) generalization. We addressed this extensively in “Section 5 - Generalization,” where we evaluated OOD generalization on the Ellipse and EllipseFlow tasks.
>
> In these experiments, we tested the model on ellipses with thicknesses both thinner and thicker than those in the training data, as well as on Reynolds numbers higher and lower than in the training distribution (see Table 2). We found that the OOD accuracy was comparable to the in-distribution accuracy. However, we observed that both types of accuracy tend to decrease as the variance of the ground-truth trajectories increases, even within in-distribution datasets. These findings are detailed in Appendix D.
>
> In the revised version, we will add results on distributional and sample accuracy for a new test case for the Ellipse and EllipseFlow tasks. Specifically, this new test dataset features an ellipse with a 10-degree orientation relative to the inflow, while the training data only included ellipses oriented parallel to it. This case further demonstrates the robust generalization capabilities of our DGN model, particularly its latent variant.
>
> **Q**: Generally, how complex a dynamical system is the model able to handle? Some dynamical systems such as the Lorenz system are chaotic, so are they still predictable with the proposed model?
>
> **A**: In principle, the proposed model can handle any dynamical system that has reached the statistical equilibrium stage. We have demonstrated its effectiveness on systems with both cyclic oscillations (e.g., the Ellipse and EllipseFlow tasks) and chaotic dynamics in statistical equilibrium (e.g., the Wing task).
>
> The Lorenz system, while chaotic and capable of exhibiting multiple attractors, still eventually reaches a statistical equilibrium where the dynamics oscillate around these attractors. A diffusion model, conditioned on the parameters of the system, could learn to generate likely states and the corresponding probability density function (PDF) of these states, much like the approach we used for the Wing task.
>
> However, we believe the Wing task presents a greater challenge than the Lorenz system. In addition to chaotic dynamics, the Wing task involves a multi-scale nature, where a wide range of interacting scales exists. Furthermore, while the Lorenz system is a three-dimensional system, the Wing task involves a much higher-dimensional state space (specifically, a |V|-dimensional space), making the modeling and prediction task more demanding.
>
> We will point out this challenge in the revised version of the paper.

---

### Official Review · Reviewer_j9He · 2024-10-31

**Soundness:** 3
**Presentation:** 4
**Contribution:** 4
**Rating:** 8
**Confidence:** 3

**Summary:**

This paper uses Denoising Diffusion to sample states from the equilibrium distribution of fluid flow tasks. Instead of operating on a 2D grid, it employs a graph-based approach combined with a multi-scale Graph Neural Network (GNN) to process information. Using only short target trajectories, the model can learn the full distribution of solutions. Additionally, by combining the diffusion approach with a variational graph auto-encoder, a latent version that operates on a reduced latent graph with fewer nodes can, in certain scenarios, improve results.

**Strengths:**

- Novelty and Innovation: The paper presents a novel and compelling approach by combining multi-scale Graph Neural Networks (GNNs), Denoising Diffusion Probabilistic Models (DDPM), and Variational Graph Auto-Encoders (VGAEs), pushing the boundaries in fluid flow modeling.

- Clear and Well-Structured Presentation: The authors provide a thorough and well-organized presentation, clearly explaining the approach and the motivation behind it, making it evident why this method is both practical and impactful.

- Relevant and Thoughtful Application: The paper demonstrates a meaningful application of the algorithm, showcasing how the method can effectively address specific challenges in fluid flow tasks.

- Strong Results and Evaluation: The results are impressive, backed by a robust evaluation that validates the approach's effectiveness and potential advantages over existing methods.

**Weaknesses:**

- Lack of Code Submission: The absence of a code submission limits the reproducibility and accessibility of the results for further research and practical implementation.

- Minor Formatting Issues: Acronyms are not consistently introduced upon first appearance, which could impact readability. For instance, "DDPM" is used without prior definition (e.g., on line 76).

- Missing Citation: The original GAN paper by Goodfellow (2014) is not cited (e.g., on line 74).

**Questions:**

- Do you plan to release the code in the future, or are there specific reasons it hasn’t been shared yet? If not, could you provide further details on any proprietary components or dependencies that may impact reproducibility? In the paper, the authors only mentioned the release of the  datasets.

---

> ### Author Response · Authors · 2024-11-19
>
> We thank the reviewer for their encouraging feedback and comments.
>
> For anonymity reasons, we have not yet made the code publicly available. However, we will release it on GitHub upon acceptance. The repository will include an installable package containing the datasets, all models and their weights, training script examples, and notebooks for demonstrating and comparing the models.
>
> The datasets will be hosted on Hugging Face for convenient access, and will be downloaded automatically via our Python package. We have also included a placeholder link at line 364 of our manuscript to indicate future code availability.
>
> We appreciate the feedback regarding formatting and citations. We have now defined acronyms like DDPM and GNN upon their first appearances and added the appropriate citation for Goodfellow’s GAN paper (2014).

---

> > ### Comment · Reviewer_j9He · 2024-11-25
> >
> > Thanks for the clarification. With that, my concerns are resolved, and I vote for the acceptance of the paper. I will maintain my score of 8.

---

### Official Review · Reviewer_praC · 2024-11-01

**Soundness:** 3
**Presentation:** 3
**Contribution:** 3
**Rating:** 8
**Confidence:** 5

**Summary:**

This paper presents two graph-based diffusion models for generating equilibrium states of fluid dynamics systems: one employing diffusion in physical space and another in the latent space generated by graph encoders. The presentation is clear and the methodological choices appear well-reasoned, particularly the use of multi-scale GNN as an encoder to support the latent diffusion model. The approach shows promise for generalization to broader applications where steady state prediction is needed.

While technically sound, the primary limitation lies in the methodology's originality. The work largely assembles existing methods rather than introducing novel techniques, though this integration is done throughoutly.

I assumed two important references should also be included and addressed to help with the literature review of this paper.

1. "Direct Prediction of Steady-State Flow Fields in Meshed Domain with Graph Networks" should be included as a baseline:
    - it directly addresses steady-state flow prediction using GNNs. This leads me to question when we should prefer deterministic versus generative approaches.
    - I can see their generative method working well when "steady states" show variations (like the periodic wakes in von Kármán vortices), but for cases with strictly stationary steady states - like this paper's ellipse and airfoil examples - wouldn't a deterministic approach make more sense?
    - I'd strongly encourage the authors to include this baseline and run comparative experiments and discuss these trade-offs based on results

2. "Efficient Learning of Mesh-Based Physical Simulation with Bi-Stride Multi-Scale Graph Neural Network," which shares similar ideas about multi-scale GNN architecture and more importantly an automatic graph hierarchy construction.
    - Frankly, compared to a refered paper in the current manuscript: (Fortunato 2022) which requires manual drawing of coarsen layer, this paper's automatic graph hierarchy construction is closer pipeline-wise. Hence I think it should be acknowledged in the related work, and the related method paragraphs.
    - While I don't think they need to conduct ablation studies here, as it's not the main focus, mentioning it would strengthen their literature review.

I recommend a borderline accept for this paper now. I would consider improving my score if the authors address these missing references, and if they add the suggested baseline and the comparative analysis between deterministic and generative approaches, and thoroughly discuss when each method works best.

**Strengths:**

- Clear writing
- The components of the whole pipeline proposed are rather reasonable
- The experiment details, the network input/output structure, and the architectures are clear and hence the reproductbility should be easy

**Weaknesses:**

- A bit lack of innovation of the components of the whole method

**Questions:**

- NA

---

> ### Author Response · Authors · 2024-11-19
>
> We are grateful for the reviewer’s constructive feedback. We will incorporate the suggested changes to improve the manuscript. In the following, we provide responses to each comment.
>
> **Q**: "Direct Prediction of Steady-State Flow Fields in Meshed Domain with Graph Networks" should be included as a baseline [...]
>
> **A**: We appreciate the reviewer’s suggestion. However, we would like to clarify that our work does not focus on predicting steady-state or mean flows, as there are already well-established deterministic methods for those tasks. Our primary goal is to learn the probability distribution function (PDF) of transient dynamics in statistical equilibrium from short trajectories, data that is too limited to represent mean flows or other statistical measures. In many applications, the primary interest lies not in the mean flow but in flow field statistics. For example, RMS fluctuations must be considered when designing airfoils or when determining the placement of aerodynamic components, and turbulence research is inherently focused on flow field statistics.
>
> While our focus is not on mean flow prediction, we agree that comparing diffusion models with direct prediction approaches for steady-state fields is an interesting direction. To address this, we will add a comparison of our DGN trained on short trajectories with two baseline models: (1) a mean-field GNN trained on the mean of the trajectories (which is limited and does not fully capture the dynamics) and (2) an expected-field GNN, which matches the expected value of the short trajectories (effectively predicting their mean). The latter model was already presented in our first submission under the name “Vanilla GNN”. As seen in Figure 4 and Tables 4 to 8, diffusion models outperform the Vanilla GNN in terms of sample and distributional accuracy. However, these metrics are less optimal for the Vanilla GNN, which only generates mean fields. To more directly address the reviewer’s concern, we will include a comparison of the diffusion models, the Vanilla GNN, and a new mean-field GNN specifically focused on their accuracy in predicting the mean. We will present these additional results in the revised manuscript.
>
> **Q**: "Efficient Learning of Mesh-Based Physical Simulation with Bi-Stride Multi-Scale Graph Neural Network," which shares similar ideas about multi-scale GNN architecture and more importantly an automatic graph hierarchy construction.
>
> **A**: Thank you for pointing this out. We will mention this in the Related Work and Methods sections.

---

> > ### Comment · Reviewer_praC · 2024-11-21
> > **reply 2nd round**
> >
> > Thank you so much for detailing the true purpose (flow characteristic distribution), agreeing to add more epxeriments, and references.
> >
> > I assume you will emphasis the purpose more in the manuscript, so that is great.
> >
> > I have 2 more concerns, further resolving them will likely improve my score to 7.
> >
> > 1. Since you have mentioned the RMS fluctuation and turbulence research, is it possible you also report the turbulence kinematic energy?
> > 2. You mentioned learning the "flow characteristics distribution" from short windows of data; and in appendix table 3, you mentioned 250 frames are used for training. Could you clarify on how these 250 frames are sampled (the leading 250 ones, or random ones); If the leading 250 ones, can you plot the turbulence kinematic energy vs time? the purpose is to see if during leading 250 frames, this value has shown enough information for the diffusion model to learn?
> >
> > Thanks for improving the manuscript,

---

> > > ### Author Response · Authors · 2024-11-22
> > >
> > > Thank you for your prompt reply and additional comments.
> > >
> > > **Q**: Since you have mentioned the RMS fluctuation and turbulence research, is it possible you also report the turbulence kinematic energy?
> > >
> > > **A**: Yes, this is a good idea. We will report the turbulence kinetic energy (TKE, $1/2(<u’u’> + <v’v’>)$) and Reynolds shear stress ($-<u’v’>$) for the test cases in the \dataset{EllipseFlow} task in a new appendix (likely Appendix D.6), alongside the mean-flow results previously requested.
> > >
> > > For the \dataset{Wing} task, we do not predict the velocity field (only the pressure field), so we cannot compute the TKE. However, the standard deviation of pressure ($\sqrt{<p’p’>}$) is already provided in Figures 1b and 6. It shows a very good match with the ground truth pressure fluctuations.
> > >
> > > **Q**: You mentioned learning the "flow characteristics distribution" from short windows of data; and in appendix table 3, you mentioned 250 frames are used for training. Could you clarify on how these 250 frames are sampled (the leading 250 ones, or random ones).
> > >
> > > **A**: Yes, the frames correspond to the first 250 consecutive frames when the data-generating simulator has reached the statistical equilibrium stage. We will clarify this in the Results section and the appendix. This approach saves time when creating the datasets, as it avoids the need to run long simulations to capture the true variance.
> > >
> > > **Q**: Can you plot the turbulence kinematic energy vs time? the purpose is to see if during leading 250 frames, this value has shown enough information for the diffusion model to learn?
> > >
> > > **A**: The reviewer is likely referring to plotting the TKE ($\frac{1}{2}(<u’u’> + <v’v’>)$) computed over increasing time intervals (for the numerical solver) or over an increasing number of samples (for the diffusion model). While we cannot provide this plot for the TKE since the task focuses exclusively on pressure, we can instead plot the pressure standard deviation ($\sqrt{<p’p’>}$).
> > >
> > > Specifically, we will add a plot of $<p’p’>(t)$, where $<p’p’>(t)$ is computed by integrating over $t_0$ to $t_0 + t$ and plotted against $t$. This would show stabilization as $t$ increases. Additionally, using the LDGN, we will create a plot of $<p’p’>(n)$ as a function of the number of samples $n$, comparing how quickly each approach stabilizes and converges to the true value.
> > >
> > > From our observations, after 250 time steps, the standard deviation obtained from the numerical solver remains far from its stable value, indicating incomplete information. This is also evident in Figures 1b and 6. In contrast, the standard deviation derived from 250 synthetic samples generated by the diffusion model is closer to convergence, and eventually reaches the expected value. We hope this addresses the reviewer’s concerns.

---

> > > > ### Comment · Reviewer_praC · 2024-11-22
> > > > **reply 3rd**
> > > >
> > > > Thank you for agreeing to add these details, which can help better reproductbility.
> > > >
> > > > As my concerns are mostly resolved (assuming these revisions will be done by authors); I am raising my score to 7.

---

### Official Review · Reviewer_5xrZ · 2024-11-03

**Soundness:** 4
**Presentation:** 2
**Contribution:** 3
**Rating:** 8
**Confidence:** 3

**Summary:**

For many fluid dynamics problems, the exact trajectory dynamics are not relevant; instead, domain scientists are more concerned with quantities of interest that arise from steady state distributions. The paper proposes a multi-scale GNN architecture contextualized within the DDPM framework. Similar to prior and current state-of-the-art diffusion models, the work uses a latent representation, instead of performing diffusion over the original data space. In order to differentiate between both contributions, experiments are performed on both DGNs (Diffusion Graph Networks) and LDGNs (Latent Diffusion Graph Networks).

Experiments are performed on 3 experimental settings: 1) pressure on the wall of an ellipse in 2D laminar flow 2) velocity and pressure around the ellipse 3) pressure on a 3D wing. The baselines include a Bayesian GNN, a Gaussian mixture model, a Gaussian regression model, and a VAE.

**Strengths:**

- The studied problem is of particular interest to the broader physics-inspired ML community and domain scientists.
- Both DGN and LDGN clearly outperform the tested baselines, and show promising results.
- The mathematical and architectural details are well-written.
- To my knowledge, this is one of the first works using diffusion models + GNNs in the space of ML4PDEs.

**Weaknesses:**

- The compared baselines are not widely used or state-of-the-art methods. At the same time, the three datasets, while representative or real-world tasks, are newly generated. This makes the results within the paper hard to contextualize within the broader literature.
- The related works section on DDPMs for graph structure needs to further differentiate the proposed method from existing works.

**Questions:**

- How does the proposed method differentiate itself from [1] (cited in the related works) which also uses a UNet style GNN? Similarly, how is the proposed GNN different / improved upon from [2]?
- Are performance improvements over the baselines a result of the diffusion model formulation or the GNN design? Have the authors considered using a latent DiT or different sampling approach (e.g., DDIM)?

[1] Haomin Wen, Youfang Lin, Yutong Xia, Huaiyu Wan, Qingsong Wen, Roger Zimmermann, & Yuxuan Liang. (2024). DiffSTG: Probabilistic Spatio-Temporal Graph Forecasting with Denoising Diffusion Models.

[2] Lino, M., Fotiadis, S., Bharath, A., & Cantwell, C. (2022). Multi-scale rotation-equivariant graph neural networks for unsteady Eulerian fluid dynamics. Physics of Fluids, 34(8), 087110.

---

> ### Author Response · Authors · 2024-11-19
>
> We appreciate the reviewer’s comments and suggestions, which will help enhance the quality and clarity of the paper. Our responses to the reviewer’s concerns are provided below.
>
> **Q**: The compared baselines are not widely used or state-of-the-art methods. At the same time, the three datasets, while representative of real-world tasks, are newly generated. This makes the results within the paper hard to contextualize within the broader literature.
>
> **A**: Most ML work in computational fluid dynamics focuses on deterministic forward prediction, whereas our work addresses the less studied but important problem of modeling distributions. This focus on probabilistic modeling makes it challenging to compare directly with commonly used datasets and baselines, as these are typically designed for solving different types of problems. Additionally, traditional CFD often relies on mesh-based data due to its ability to model complex geometries, such as 3D wings. In contrast, many ML models use grid-based data for compatibility with CNNs. Hence, we compared DGNs with deterministic multi-scale GNNs (the ‘vanilla’ GNN).
>
> To ensure a fair comparison, we also selected representative probabilistic modeling approaches commonly used in fluid dynamics research. Specifically, Maulik et al. utilized Gaussian Mixture Models (GMMs) for uncertainty modeling, while Liu et al. applied Gaussian Regression and Bayesian networks. Variational Autoencoders (VAEs) were also chosen as a baseline given their popularity in generative modeling across domains.
>
> Regarding the datasets, while some mesh data is available [Pfaff 2021], to the best of our knowledge, no other studies have addressed large-scale, unsteady, and turbulent dynamics on detailed unstructured meshes, as in our Wing dataset. This justified our use of existing data for canonical cases (Ellipse and EllipseFlow) and motivated the generation of the new Wing dataset to provide a real-world application context. Creating and publishing this dataset is itself a valuable contribution, and we will emphasize this in Appendix C.

---

> ### Author Response · Authors · 2024-11-19
>
> **Q**: The related works section on DDPMs for graph structure needs to further differentiate the proposed method from existing works.
>
> **A**: We will expand the related works section to clarify that, to our knowledge, this is the first application of graph-based DDPMs to learn the full probability distribution from partial trajectories. Also, we will add the points mentioned in the point right below.
>
> **Q**: How does the proposed method differentiate itself from [1] (cited in the related works) which also uses a UNet style GNN? Similarly, how is the proposed GNN different / improved upon from [2]?
>
> **A**: In [1], the nodes are processed in a flat structure, not a hierarchy, and the term U-Net refers to the time dimension of the feature vectors. Specifically, this model performs conventional pooling and unpooling in the temporal dimension (compressing the feature vectors) rather than in the spatial dimension (graph compression). This approach maintains the same graph structure and node count in the latent space, leading to a high cost for message-passing operations. As a result, this method is constrained to small graphs (34, 41, and 170 nodes in their datasets).
>
> In contrast, our approach employs a multi-scale U-Net GNN with hierarchical graph compression. We leverage a multigrid mesh coarsening algorithm, and message passing between high-resolution and low-resolution graphs. Graph compression is significantly more challenging than feature compression due to the unstructured nature of the data, but it provides substantial benefits, as demonstrated in Appendix D.2. To further illustrate this point, we are adding additional results to Appendix D.2., showing that a flat GNN fails to learn effectively on the large-scale Wing task, highlighting the advantages of our hierarchical approach. We will clarify this in the related work to strengthen the differences with existing studies.
>
> As noted by Reviewer PraC, ours is not the only GNN with a hierarchical graph structure. Other studies, such as Fortunato et al. and “Efficient Learning of Mesh-Based Physical Simulation with Bi-Stride Multi-Scale Graph Neural Network,” also employ graph hierarchies. However, the approach in [2] is inefficient for our Wing task, since coarsening relies on a voxel grid whose nodes and edges do not lie in the Wing surface. Instead, the multigrid coarsening that we employed guarantees that the coarser nodes and edges are still located on the Wing surface. This avoids undesired edges connecting nodes that are spatially closed in the world-space, but many hops away in the mesh-space (e.g., a pair of nodes in the upper and lower Wing surfaces). We will clarify this in Appendix B.
>
> **Q**: Are performance improvements over the baselines a result of the diffusion model formulation or the GNN design? Have the authors considered using a latent DiT or different sampling approach (e.g., DDIM)?
>
> **A**: Our performance improvements stem from both the multi-scale GNN design and the diffusion model formulation. We study the effect of these contributions individually in the “Multi-scale vs. single-scale DGN” Section and “Comparison to probabilistic baselines” Section of our paper.
>
> Due to their data-intensive nature, we have not yet considered transformer architectures, though adding linear attention in the latent space is a promising area for future work (we will mention this in the conclusions). For sampling, we opted for the approach from Nichol et al. (2021) because in their work it proved to outperform DDIM in sample accuracy for 50+ denoising steps, although we did not compare them in our research.
>
> Flow matching is a recent generative model that shares many aspects with diffusion models, while allowing for fewer denoising steps during sampling. We will add an appendinx with preliminary comparisons between DGN, LDGN, Flow Matching (FM), and latent FM models on the EllipseInDist dataset, across various denoising step counts. Early tests show that with fewer than 10 denoising steps, FM models outperform DGN models, while for 50 steps, diffusion models maintain an accuracy advantage over FM. We believe our results indicate that the proposed multi-scale approach will be applicable to a broad range of probabilistic diffusion models
>
> [Lipman] Lipman, Y., Chen, R.T., Ben-Hamu, H., Nickel, M. and Le, M., 2022. Flow matching for generative modeling. arXiv:2210.02747.

---

> > ### Comment · Reviewer_5xrZ · 2024-11-20
> > **Reply to the Authors**
> >
> > Thank you taking the time to answer my questions and updating the manuscript. In light of the new manuscript, I have chosen to raise my score 6 $\rightarrow$ 8.

---

### Author Response · Authors · 2024-11-19
**General reply to all reviewers**

We thank the reviewers for their thoughtful feedback, which we are incorporating to improve the completeness and clarity of our manuscript. These modifications and additions are detailed in our replies to each reviewer. Below, we highlight the most significant updates:

- We have clarified the differences between our method and prior work in the Related Work section and Appendix B.
- We have clarified the concept of statistical equilibrium and provided a precise definition of the learning problem in the manuscript.
- To further demonstrate the generality of our method, we include performance results for a new test dataset for the Ellipse and EllipseFlow tasks. This dataset features an ellipse with a 10-degree angle relative to the free stream, whereas the training data assumes a 0-degree angle.
- We justify the choice of our sampling approach and introduce Flow Matching as an alternative for faster inference. We will add an appendix with preliminary comparisons between DGN, LDGN, Flow Matching (FM), and Latent FM (LFM) models on the EllipseInDist dataset. Our findings show that the proposed method naturally extends to Flow Matching, and yields accurate samples with fewer sampling steps (10 for this scenario).
- Reviewer praC raised concerns about the lack of a deterministic baseline trained specifically on the mean flow. To address this, we will include an additional baseline and compare its performance with the DGN and the “vanilla GNN”. The latter already represents a deterministic GNN trained to match the expectation of states observed during training.
- In the original Results section, we hypothesized that the success of diffusion-based models in learning the full distribution from short trajectories stems from their ability to identify shared physical patterns across systems. To strengthen this claim, we conducted a new experiment to assess model performance when trained on datasets with varying numbers of short trajectories. Results show that DGN and LDGN outperform baseline models, suggesting that these models effectively interpolate among trajectories, whereas baseline models rely more heavily on having seen specific data.
- We have updated the GPU runtime for PDF estimation in Table 10. The revised measurement was taken with nearly 100% GPU utilization, unlike the previous measurement, which did not fully utilize available GPU power. CPU and single-sample-generation runtime remain unchanged.

We will post an update once the revised manuscript has been uploaded.

---

> ### Author Response · Authors · 2024-11-25
> **Paper updated**
>
> Once again, we would like to thank all the reviewers for their valuable questions and comments. These have been carefully addressed in the revised version of our manuscript, as outlined in our previous general comment and individual replies. The updated version of the manuscript is now available.
>
> We have also included several new appendices to support and extend some of our arguments:
>
> - Appendix C: We provide an explanation of incompressible flow and its associated challenges in relation to our objectives and datasets.
> - Appendix D.5: We evaluate model performance when trained on datasets with varying numbers of short trajectories. The results demonstrate that DGN and LDGN outperform baseline models, suggesting their effectiveness in interpolating among trajectories, whereas the baseline models are more reliant on having seen specific data.
> - Appendix D.6: (In response to Reviewer praC) We present results showing that diffusion-based models outperform probabilistic and deterministic baselines in learning the mean flow from incomplete simulations, as well as in predicting the TKE and the Reynolds shear stress (which are computed from their learned distributions).
> - Appendix D.7: We introduce flow-matching in our learned latent space as an alternative approach requiring fewer transition steps than denoising diffusion.
>
> Regarding Reviewer j9He, due to space limitations in the main paper, we have included dummy links to our code and datasets at the beginning of Appendices B and C, respectively.
>
> All changes have been highlighted in red in this revised version of the paper.

---

### Meta-Review · Area_Chair_cRgx · 2024-12-20

**Metareview:**

This paper on computational fluid dynamics proposes a multi-scale GNN architecture combined with diffusion models working both in physical and in latent space. Five expert reviewers provided very favorable reviews and appreciated novelty, technical soundness, excellent performance, generalization capabilities, an well-written paper, strong evaluation, good discussions. The authors could answer the questions by the reviewers, and there the assessments were consistent and positive. The AC concurs and recommends acceptance.

**Additional Comments On Reviewer Discussion:**

The reviewers engaged with the authors.

---

### Decision · Program_Chairs · 2025-01-22

Accept (Oral)